# Cardio-metabolic traits and its socioeconomic differentials among school children including metabolically obese normal weight phenotypes in India: A post-COVID baseline characteristics of LEAP-C cohort

Mani Kalaivani[1]*, Chitralok Hemraj[1], Varhlunchhungi Varhlunchhungi[1], Lakshmy Ramakrishnan[2], Sumit Malhotra[3], Sanjeev Kumar Gupta[3], Raman Kumar Marwaha[4], Ransi Ann Abraham[2], Monika Arora[5], Tina Rawal[5], Maroof Ahmad Khan[1], Aditi Sinha[6] Nikhil Tandon[7]

1 Department of Biostatistics, All India Institute of Medical Sciences, New Delhi, India, 2 Department of Cardiac-Biochemistry, All India Institute of Medical Sciences, New Delhi, India, 3 Centre for Community Medicine, All India Institute of Medical Sciences, Delhi, India, 4 International Life Sciences Institute, Delhi, India, 5 HRIDAY, Green Park Extension, Green Park, Delhi, India, 6 Department of Pediatrics, All India Institute of Medical Sciences, New Delhi, India, 7 Department of Endocrinology & Metabolism, All India Institute of Medical Sciences, Delhi, India.

* manikalaivani@gmail.com

## Abstract

### Background

Cardio-metabolic risks emerge in early life and progress further in adult life. In recent times, COVID-19 pandemic aggravated risks owing to poor food security and diet quality. We aimed to assess the prevalence of cardiometabolic traits including the metabolically obese normal weight phenotype and its socioeconomic differentials in children and adolescents aged 6–19 years in India.

### Methods

A baseline assessment was conducted between August and December, 2022, as part of a school-based cohort study that aimed at longitudinally evaluating the anthropometric and metabolic parameters among urban children and adolescents aged 6–19 years from three public (non-fee paying) and two private (fee paying) schools in India. Private and public schools were considered as a proxy for higher and lower socioeconomic status respectively. Blood pressure and blood samples in a fasting state were obtained only from adolescents aged 10–19 years. The prevalence and its 95% confidence interval using the Clopper exact method and adjusted prevalence ratios were calculated using random-effects logistic regression models.

**Data availability statement:** The data used in this study are an extract from an ongoing longitudinal study and cannot be shared publicly before the completion of the study. Interested researchers can request access to the data that support the findings of this study by contacting the corresponding author or upon reasonable request to the Institute Ethics Committee (ethicscommitteeaiims@gmail.com) after the study's completion.

**Funding:** This study was funded by the Indian Council of Medical Research (Grant No: 5/4/1-15/2020-NCD-1, awarded to MK), under the Department of Health Research, Ministry of Health and Family Welfare, Government of India. The funder had no role in study design, data collection and analysis, decision to publish, or preparation of the manuscript.

**Competing interests:** The authors have declared that no competing interests exist.

## Findings

Among 3888 recruited students, 1985 (51.05%) were from public schools, and 1903 (48.95%) were from private schools aged 6–19 years. The overall prevalence of underweight was 4.95% (95% CI 4.29–5.69), with a significantly higher prevalence in public schools (8.09%) than private schools (1.69%). The overall prevalences of general obesity and central obesity were 13.41% (95% CI 12.35–14.52) and 9.15% (95% CI 8.26–10.11), respectively, with significantly higher prevalence in private schools (p < 0.001). The prevalences of general and central obesity were four times (adjusted PR = 4.42, 95% CI 2.90–6.72) and eight times (adjusted PR = 8.31, 95% CI 4.82–14.35) higher, respectively, in private schools than public schools. The overall prevalence of hypertension was 7.37% (95% CI 6.33–8.51), and similar prevalences were found in public and private schools. Private school students had 2.37 times higher prevalence of impaired fasting plasma glucose (adjusted PR = 2.37, 95% CI 1.19–4.72) and 3.51 times higher prevalence of metabolic syndrome (adjusted PR = 3.51, 95% CI 1.54–8.01) than public school students. Among 2160 adolescents, 67.73% (1463) had normal body mass index. The prevalence of metabolically obese normal weight phenotype (MONW) was 42.86% (95% CI 40.30–45.44), which is higher in public [46.39% (95% CI 43.25–49.54)] than private [35.33% (95% CI 30.99–39.86)] schools (p < 0.001) with adjusted PR of 0.91 (95% CI 0.70–1.17). The most prevalent cardio-metabolic abnormality among metabolically obese normal weight phenotype was low high-density lipoprotein-c, significantly higher among adolescents from public schools (62.12% vs 52.73%, p = 0.039) than private schools. The prevalence of metabolically obese underweight (MOUW) (48/115) was 41.74% (95% CI 32.61–51.30), being higher among adolescents in public schools than private schools but not significant (p = 0.264).

## Interpretation

Effective implementation of food security measures and targeted initiatives will be crucial to mitigate the socioeconomic disparities associated with the growing burden of cardiometabolic traits. Metabolic obesity among phenotypically normal or underweight adolescents should not be overlooked but should be intervened early through novel screening criteria to prevent future cardiovascular burdens. These findings also have implications for low- and -middle income countries like India, which are undergoing a nutritional transition where socioeconomic status strongly influences cardio-metabolic traits.

## Introduction

Currently, the double burden of underweight and obesity in children and adolescents is a global public health challenge. Globally, 190 million and more than 390 million children and adolescents aged 5–19 years were underweight and overweight (with

160 million of them classified as obese) respectively [1]. According to World Health Organization (WHO) criteria, the prevalences of underweight, overweight and obesity in low-and-middle income countries were 4.7%, 17.3% and 8.6%, respectively, among adolescents aged 12–15 years [2]. This ongoing obesity epidemic urges the importance of understanding body composition for short-term and long-term health. The main component of body composition having cardio-metabolic implications is body fat, and excess accumulation of body fat can lead to physical health problems, psychosocial issues and long-term health consequences [3–7].

Overweight and obese children and adolescents are identified using conventional growth charts. These are usually derived from cross-sectional data, which examine an individual at a single point in time. In contrast, longitudinal data of an individual with multiple measurement are more informative than a single measurement in studying the growth path. Understanding an individual's prior growth pattern gives us a clearer picture of his/her current growth path. The longitudinal growth charts are important during infancy and puberty as the individuals on the upper (or lower) centiles tend to move toward the median more rapidly than others. During these stages, conventional growth charts may give us an incorrect impression of abnormal growth [8,9]. Globally, studies reporting longitudinal reference data for growth velocity in children are available [10–13]. However, in India, we have a growth chart constructed from cross-sectional data for some of the anthropometric parameters like height, weight, body mass index (BMI), percentage body fat, waist circumference, waist-hip circumference ratio and triceps and subscapular skinfold thicknesses [14–18], but the longitudinal growth reference data are relatively absent. The emerging data on childhood obesity in developing countries urges the need to monitor secular trends in growth parameters for children and adolescents [19] to enable future projection of overweight, obesity, and their downstream metabolic perturbations such as dysglycemia, dyslipidemia and insulin resistance. To address these, the Longitudinal Evaluation of Anthro-metabolic Parameters in Children (LEAP-C) study was designed, (i) to develop longitudinal age and sex-specific percentiles of anthropometric measures, blood pressure, serum lipids and fasting insulin and plasma glucose levels for adolescents, (ii) to formulate longitudinal reference for growth velocity in children and adolescents and (iii) to correlate changes in anthropometric measures with blood pressure, serum lipids, fasting insulin and plasma glucose levels over a period of three years in adolescents.

In India, data on the impact of the COVID-19 pandemic on obesity are scarce, and there are even fewer data on underweight [20]. Also, there is a regional disparity in the rates of overweight (2.28%-21.90%) and obesity (2.40%-17.60%) pre-COVID among school-going children and adolescents [21]. Studies used different criteria derived from the Western population for screening overweight/obesity, which might affect the magnitude of the prevalence. These criteria include the World Health Organization (WHO) [22], the International Obesity Task Force (IOTF) [23–25], the United States Centers for Disease Control and Prevention (CDC) growth reference [26] and the Indian Academy of Pediatrics (IAP) [27]. Therefore, reporting prevalences using all these criteria within a population is imperative to enable a more accurate comparison. No studies were found in the literature, reporting comprehensively encompassing commonly used criteria, the prevalence of underweight, overweight and obesity among children and adolescents covering the broad spectrum of age groups.

Though there is abundant literature on the prevalence of overweight/obesity along with its variation in different socio-economic status (SES) among Indian adolescents, there is a significant gap in the comprehensive evaluation of their cardio-metabolic health. It is particularly concerning that, in India, every fifth person is an adolescent aged 10–19 years, and this group is vulnerable to the dual burden of under and overnutrition [28]. This highlights the need for a deeper understanding of how these risks are distributed across different SES and to tailor the population's needs through effective policies and programs. Next, BMI is a simple and widely used index to classify general obesity; however, it does not distinguish between lean and fat tissue [29]. It has been recognized that individuals with normal weight by BMI could also have high cardio-metabolic risk [30]. Little information is known about these risks among Indian adolescents. Therefore, this study addresses from the baseline assessment of LEAP-C (i) the prevalence of anthropometric indicators-general obesity and central obesity among children and adolescents, (ii) cardio-metabolic traits - childhood hypertension, lipid abnormalities (dyslipidemia), elevated fasting plasma glucose and fasting insulin, metabolic syndrome (MetS) and metabolically

obese normal weight (MONW) phenotype among adolescents between public and private school going urban students in post-COVID era in India.

## Materials and methods

### Study design

LEAP-C study is a longitudinal school-based study. The study flow is shown in Fig 1.

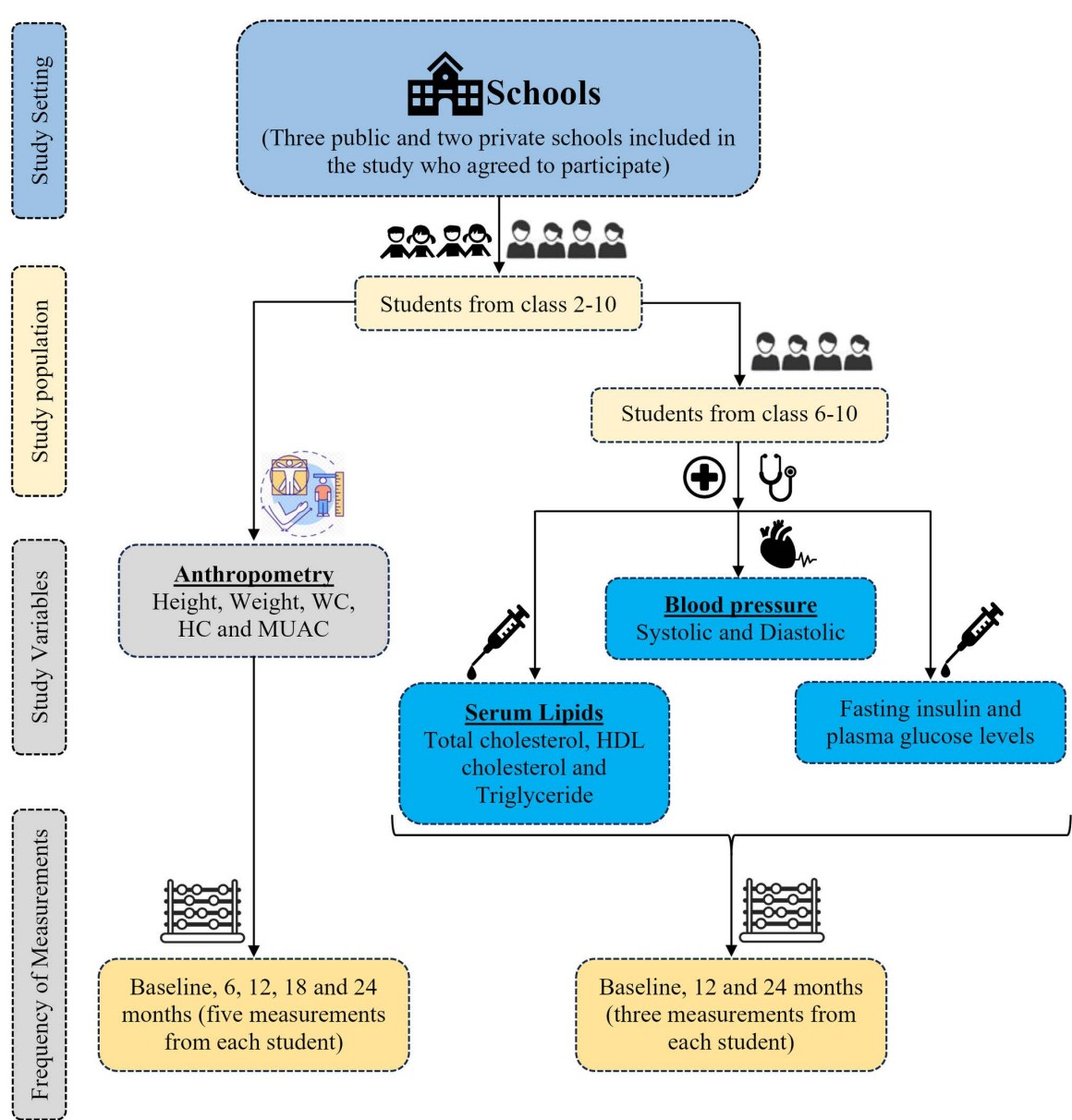

**Fig 1. Flow chart with details of study setting, population, variables and frequency of measurements.** (WC: Waist Circumference, HC: Hip Circumference, MUAC: Mid Upper Arm Circumference and HDL-c: High-Density Lipoprotein-cholesterol).

## Recruitment strategy

The study comprised five schools in Delhi, India, including three public and two private schools (convenient selection as schools were reluctant due to the pandemic). Private schools recruited in the study were based on the approval received from respective school principals. For public schools, approval was sought from the Director (Education), New Delhi Municipal Council, New Delhi.

The study was conducted between August 17, 2022, and December 20, 2022; the study participants were children and adolescents aged 6–19 years. The school type, public (non-fee paying) and private (fee paying), was used as a proxy variable for SES as done in other studies [31–33], public representing low SES and private representing high SES. The study team met with each school principal to elaborate on the study objectives, expected outcomes and its impact to establish rapport and trust. Informed written consent was obtained from the parents/guardians of all participating students from classes 2–10 at the initiation of the study. Verbal assent (in the presence of coordinating teacher and research staff) from participating students of classes 2–5 and written assent from participating students of classes 6–10 were also obtained.

## Inclusion and exclusion Criteria

All students studying in the 2nd to 10th class of public and private schools who gave their assent were included in the present study.

Children with known history (self-reported) of chronic systemic illness (such as tuberculosis, liver disease, renal disease and other diseases likely to affect normal growth), major physical deformity (e.g., cerebral palsy, musculoskeletal disorders), those on medication for chronic illnesses, any condition precluding anthropometric measurements (e.g., knee injury, leg fracture, etc.) at the time of assessment, history of conditions treated with medications (steroids, β-blockers, diuretics, etc.), associated with dyslipidemia (Cushing's disease, renal disorders, etc.) and thyroid dysfunction were excluded from the study.

## Ethical consideration and approval

The study protocol was approved (IEC-354/08.05.2020, RP-31/2021) by the Institute Ethics Committee of All India Institute of Medical Sciences, Delhi, India. The study had a meaningful engagement with key stakeholders such as principals, teachers and parents. Individualized measurement reports were sealed in envelopes and distributed to all participants through their schools. Parents of participants with abnormal findings were informed about the importance of physical activity, a balanced diet, and reduced screen time to support the children's well-being during the parent-teacher meeting. The team had no direct role in counselling, as the school authorities did not permit them to interact with the participants' parents. However, the team offered medical support to parents who requested assistance for their children.

## Non-respondents

The non-respondents included children who did not have their parents' consent or did not give written assent, children who were absent, and those who reported for blood sampling in a non-fasting state after two attempted visits during baseline assessment.

## Discontinuation criteria

Students were allowed to withdraw at any time point during the study and reasons for withdrawal were taken from each student who opted to withdraw from the study.

## Timing of assessments

The baseline assessment occurred between August 17, 2022, and December 20, 2022. Longitudinal follow-up assessments of students were conducted semi-annually for anthropometric parameters in children and adolescents studying in

classes 2nd to 10th, whereas biochemical parameters and blood pressure in adolescents studying in classes 6th to 10th were measured annually over a three-year period. The last follow-up of the study is yet to be completed.

## Anthropometric, blood pressure and biochemical measurements

Anthropometric parameters included height (cm), weight (kg), waist circumference (WC in cm), hip circumference (HC in cm) and mid-upper arm circumference (MUAC in cm). All participants were prepared for the anthropometric measurement by removing any extra clothing, ornamental items and shoes. All measurements were taken in separate rooms for boys and girls to ensure privacy and comfort for the participants. Height was measured to the nearest 0.1 centimetre using a portable stadiometer with the participant facing the assessor, standing straight, arms on the side, feet together, heels touching the backboard, knee straight and head held in Frankfort horizontal plane. Weight was measured to the nearest 0.1 kilogram using a digital weighing scale with the participant facing forward, standing straight and with arms on the side. WC, HC and MUAC were measured to the nearest 0.1 centimetre with stretch-resistant tape. WC measurement was taken at the midpoint of the inferior margin (lowest point) of the last rib and the iliac crest (top of the hip bone). HC measurement was taken at the maximum circumference of the buttock. MUAC was taken at the midpoint of the acromion process of the scapula and tip of the olecranon process (the bony part of the mid-elbow) [34].

The systolic blood pressure (SBP in mmHg) and diastolic blood pressure (DBP in mmHg) were measured by trained staff in a separate room using OMRAN automatic oscillometer device (OMRON) with small, medium and large cuffs. After resting for five minutes, three readings were taken, and the average of the closest two measurements was used [35].

A 5 ml of blood was collected from each participant who had fasted overnight by trained phlebotomists. The blood was collected under sterile conditions and stored at an appropriate temperature before being analyzed. The blood sample was then analyzed to measure serum lipids (total cholesterol (TC in mg/dl), serum triglycerides (TG in mg/dl), and high-density lipoprotein cholesterol (HDL-c in mg/dl)), fasting insulin (µIU/ml) and fasting plasma glucose (FPG in mg/dl) levels.

## Quality assurance and quality control strategies

The investigators developed standard operating procedures (SOPs) for anthropometry, blood pressure, and blood collection, and staff were trained to follow them. Field staff received training before baseline and retraining for follow-up assessments. Five assessors (two males for boys, and three females for girls) conducted the measurements. Weighing scales and stadiometers were calibrated before each assessment, and the oscillometric device was regularly checked against a mercury sphygmomanometer. Staff were also trained before each visit for anthropometry and blood pressure assessments.

Biochemical parameters were quality assured at two levels: internal quality control (IQC) and external quality assurance. At level 1, the laboratory followed internal SOPs for accuracy, precision, and reproducibility. At level 2, external quality control was implemented with support from the Randox International Quality Assurance Scheme (RIQAS), UK.

## Operational definitions

The operational definitions used to define anthropometric indicators and cardio-metabolic traits are given in the S1 file.

## Sample Size

LEAP-C study was planned to establish the reference values of height, weight, BMI, waist circumference, hip circumference and mid-upper arm circumference in children aged 7–15 years, and blood pressure, serum lipids, fasting insulin and plasma glucose levels in children aged 11–15 years. The longitudinal reference for each age will be provided by calculating centile as this involves complicated procedures and equations demanding a higher sample size to estimate extreme centiles, and account for increased variability during the pubertal period and smoothing of the curve [36,37]. A study found

that comparing 20,000 measurements for each sex from birth to 20 years with 2,000 measurements showed little difference in the Lambda, Mu, and Sigma curves used to calculate centiles [38]. Therefore, 100 children in each age group would be sufficient to establish reference centiles.

A sample size of 100 subjects at each age and sex will be included to assess growth references and height velocity during puberty with measurements taken every six months for two and a half years. Also, children under seven and over 15 years will be sampled to address greater standard error at extreme ages. Blood sampling/blood pressure measurements were performed only for children aged 11–15 years. However, the measures applied above do not address biochemical parameters. The sample size was increased to 110 children in each age group and sex to account for 10% refusals and retention. Finally, the total sample size of 1980 students aged 6–15 years from public and private schools for anthropometry parameters and as the focus of the study for blood pressure/biochemical parameters was only among 11–15 years, a subset of 1100 students were considered.

### Data management and statistical analysis

LEAP-C study data were entered and monitored using the Research Electronic Data Capture (REDCap) hosted at All India Institute of Medical Sciences, Delhi, India. The baseline profile of the study participants is summarized as mean±SD (continuous data) and frequency (%) for categorical data. Anthropometric measurements, blood pressure and biochemical parameters were compared between public and private schools and boys and girls using Student's t-test for independent samples. The prevalences of cardio-metabolic traits between public and private schools among boys, girls and overall were estimated to examine the variations by SES. Additionally, we have estimated prevalence of obesity using three different criteria-WHO, IOTF and IAP – separately for boys and girls in both public and private as well as for the total sample. We analyzed age-related patterns by segregating age into three age groups: children (6–9 years), early adolescents (10–14 years), and late adolescents (15–19 years).

All prevalence estimates were reported with a 95% confidence interval using the Clopper-Pearson exact method, using the *"prop"* command in Stata 18.0 (StataCorp LP, USA). For missing data on parental education, Multiple Imputation by Chained Equations (MICE) was used with a proportional odds logistic model employing 10 imputations and 20 repetitions of each imputation. The imputed data with the distribution closest to that of the original data were used for analysis. Random effects logistic regression with marginal standardization accounting for clustering of schools was done to examine the influence of SES (school type) differential on various cardio-metabolic traits, including MONW phenotype adjusting for children's age and gender, and their father's and mother's education to estimate prevalence ratios (PR) using R 4.3.3 (R Foundation of Statistical Computing, Vienna, Austria) *"prLogisticDelta"* package. All estimates were reported using 95% CI. The same analysis was done on gender-segregated data to explore whether SES would be associated with cardio-metabolic traits. All p-values less than 0.05 were considered statistically significant.

## Results

### Demographic profile

Of 2142 public school and 2914 private school students from 2nd to 10th class, 99.63% and 99.62%, respectively, were eligible for the study. The response rate in public schools was 93.02% (n = 1985) and in private schools was 65.55% (n = 1903) in the baseline assessment of the LEAP-C study. Among the study participants, 31.56% were children (6–9 years) and 68.44% were adolescents (10–19 years). Of the 3888 recruited students of classes 2nd to 10th, 99.18% (3856) had anthropometric data. Among the 2661 students from classes 6–10, 85.72% (2281) had blood pressure, and 82.56% (2197) had biochemical data. The difference in mean ages between public and private school students was not statistically significant among children (8.48 ± 0.95 vs 8.38 ± 0.93, p = 0.060) and significant among adolescents (13.30 ± 1.85 vs 12.93 ± 1.74, p < 0.001). Among the parents, 22.01% of fathers of public school children and adolescents aged 6–19 years

had no formal education compared to 3.45% in private schools, with a p-value of <0.001. Similarly, a significantly higher proportion of mothers (36.13%) of public school children and adolescents aged 6–19 years had no formal education compared to 2.64% of mothers of private schools with p<0.001 (Table 1).

### Anthropometric, cardio-metabolic and biochemical profile

The anthropometric profile (S1 Table) of children and adolescents and the clinical and metabolic profiles (S2 Table) of adolescents, were compared between public and private schools, and between boys and girls. The mean of all the anthropometric parameters, systolic and diastolic blood pressures and biochemical parameters were significantly higher in private school students compared to public school students.

When comparing boys and girls, the mean BMI and HC did not significantly differ (BMI: p=0.751 and HC: p=0.056) among school-going children aged 6–19 years. SBP was higher in boys (p<0.001), while DBP was higher in girls (p<0.001) compared to their counterparts. Among lipid parameters, all means were significantly higher (except HDL-c, p=0.168) in girls than boys.

**Table 1. Demographic characteristics of study participants (n=3888).**

| Characteristics | Before Imputation | | | | After Imputation | | | |
|---|---|---|---|---|---|---|---|---|
| | Total | Public | Private | p-value | Total | Public | Private | p-value |
| **Age Groups** | 3888 | 12.08±2.68 | 11.20±2.66 | <0.001 | | | | |
| **Children (6–9 years)** | 1227 | 8.48±0.95 | 8.38±0.93 | 0.060 | | | | |
| Boys | 657 | 276 (54.87) | 381 (52.62) | 0.438 | | | | |
| Girls | 570 | 227 (45.13) | 343 (47.38) | | | | | |
| **Early Adolescents (10–14 years)** | 2162 | 12.63±1.41 | 12.44±1.44 | 0.002 | | | | |
| Boys | 1264 | 691 (59.01) | 573 (57.82) | 0.576 | | | | |
| Girls | 898 | 480 (40.99) | 418 (42.18) | | | | | |
| **Late Adolescents (15–19 years)** | 499 | 15.83±0.82 | 15.50±0.38 | <0.001 | | | | |
| Boys | 313 | 199 (63.99) | 114 (60.64) | 0.453 | | | | |
| Girls | 186 | 112 (36.01) | 74 (39.36) | | | | | |
| **Father Education** | 3560 | 1781 | 1779 | | 3888 | 1985 | 1903 | |
| No formal education | 454 | 392 (22.01) | 62 (3.49) | <0.001 | 497 | 422 (21.26) | 75 (3.94) | <0.001 |
| Primary | 302 | 298 (16.73) | 4 (0.22) | | 335 | 320 (16.12) | 15 (0.79) | |
| Middle | 389 | 376 (21.11) | 13 (0.73) | | 425 | 398 (20.05) | 27 (1.42) | |
| Secondary/High | 474 | 398 (22.35) | 76 (4.27) | | 519 | 437 (22.02) | 82 (4.31) | |
| Intermediate/Diploma | 418 | 238 (13.36) | 180 (10.12) | | 461 | 263 (13.25) | 198 (10.40) | |
| Graduation | 926 | 71 (3.99) | 855 (48.06) | | 1006 | 105 (5.29) | 901 (47.35) | |
| PG & above | 597 | 8 (0.45) | 589 (33.11) | | 645 | 40 (2.02) | 605 (31.79) | |
| **Mother Education** | 3554 | 1777 | 1777 | | 3888 | 1985 | 1903 | |
| No formal education | 689 | 642 (36.13) | 47 (2.64) | <0.001 | 752 | 678 (34.16) | 74 (3.89) | <0.001 |
| Primary | 449 | 433 (24.37) | 16 (0.90) | | 488 | 459 (23.12) | 29 (1.52) | |
| Middle | 328 | 298 (16.77) | 30 (1.69) | | 360 | 317 (15.97) | 43 (2.26) | |
| Secondary/High | 330 | 242 (13.62) | 88 (4.95) | | 366 | 270 (13.60) | 96 (5.04) | |
| Intermediate/Diploma | 291 | 110 (6.19) | 181 (10.19) | | 325 | 130 (6.55) | 195 (10.25) | |
| Graduation | 848 | 47 (2.64) | 801 (45.08) | | 921 | 82 (4.13) | 839 (44.09) | |
| PG & above | 619 | 5 (0.28) | 614 (34.55) | | 676 | 49 (2.47) | 627 (32.95) | |

*Data are presented as mean±SD and number (%).*

None of the lipid parameters in private school students showed significant differences between boys and girls. In public school students, however, the mean of all the lipid parameters (except HDL-c, p = 0.792) was higher among girls than boys.

### Prevalence of underweight

According to the IAP criteria, the prevalence of underweight was 7.75% (95% CI 5.57–10.45) in children, 8.19% (95% CI 6.68–9.92) in early adolescents and 8.25% (95% CI 5.41–11.94) in late adolescents in public schools, which was higher compared to private schools. Using the IOTF criteria, the prevalence of underweight among children, early and late adolescents, was approximately five times higher in public schools than the prevalence observed with the IAP criteria (S3 Table and Fig 2).

According to all three criteria, among children aged 6–9 years, the prevalence of underweight is higher in girls than boys in public schools (WHO: p = 0.348, IOTF: p = 0.107 and IAP: p = 0.567) whereas in early (WHO: p = 0.008, IOTF: p = 0.116 and IAP: p = 0.049) and late adolescents (WHO: p = 0.321, IOTF: p = 0.686 and IAP: p = 0.658), underweight prevalence is higher in boys than girls (p-value not shown in Figs 3 and 4). According to the IOTF criteria, the prevalence of underweight is notably higher across all age groups for both boys and girls in public and private schools compared to other criteria. According to the IAP criteria, in private schools, the prevalence of underweight is higher among girls than boys across all age groups.

### Prevalence of overweight/obesity

The prevalence of obesity (S3 Table and Fig 2) is the lowest according to IOTF criteria followed by WHO and IAP criteria among children and adolescents aged 6–19 years. In private schools, the prevalence of obesity according to IAP criteria decreases with age, with rates of 24.17% (95% CI 21.09–27.46) in children, 23.45% (95% CI 20.83–26.23) in early adolescents and 12.97% (95% CI 8.49–18.69) in late adolescents. This decreasing trend with age was not observed in public schools, where the obesity prevalence remained relatively stable across age groups: 3.98% (95% CI 2.45–6.07) in children, 4.66% (95% CI 3.52–6.03) in early adolescents and 4.62% (95% CI 2.55–7.63) in late adolescents (S3 Table and Fig 2).

Noticeably, the prevalence of obesity was higher among boys than girls in both types of schools according to all three criteria. By the IAP criteria, the prevalence of obesity was higher in boys aged 6–9 years, 29.13% (95% CI 24.62–33.98), followed by early adolescents, 27.64% (95% CI 24.00–31.52) and late adolescents, 16.67% (95% CI 10.34–24.80) in private schools. Moreover, the prevalence of obesity is significantly higher in private schools than in public schools. The combined prevalence of overweight and obesity in children and adolescents aged 6–19 years is nearly 4 times higher in private schools (46.72% vs 12.11%) than in public schools with p < 0.001.

### Prevalence of central obesity

Like general obesity, the prevalence of central obesity among children and adolescents aged 6–19 years was significantly higher in private schools at 16.77% (95% CI 15.11–18.53) compared to public schools at 1.83% (95% CI 1.29–2.53) with adjusted PR of 8.31 (95% CI 4.82–14.35) (Table 2 and Fig 5). A similar pattern was observed between private and public school students in both boys and girls. However, the prevalence of central obesity was very low among public school girls, 0.61% (95% CI 0.20–1.43).

### Prevalence of hypertension

The prevalence of hypertension was 7.37% (95% CI 6.33–8.51) among urban adolescents aged 10–19 years (Table 2). There is no difference in the prevalence of hypertension (Tables 3 and 4) among adolescents between public and private schools and between boys (p = 0.842) and girls (p = 0.626).

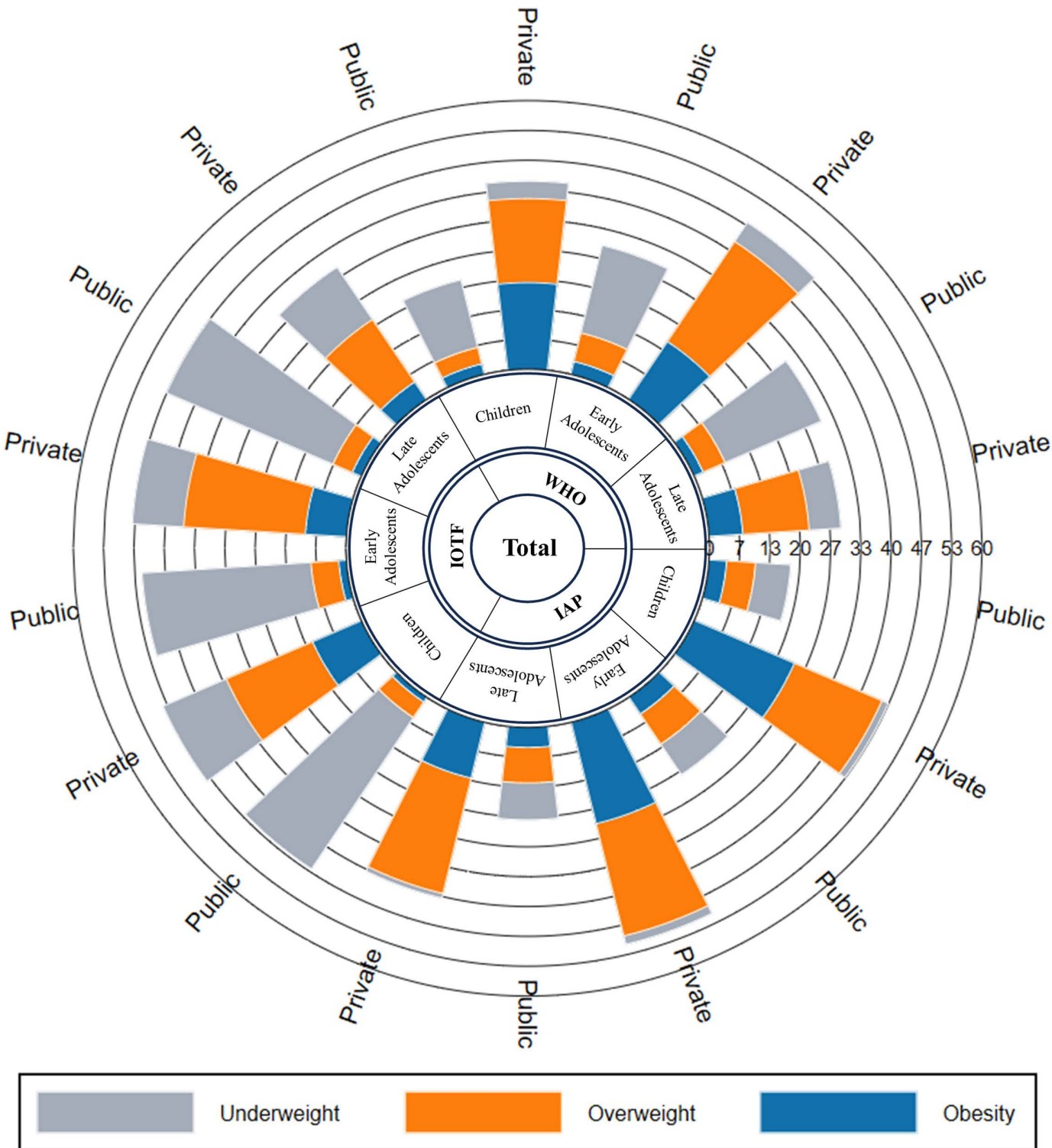

**Fig 2. Prevalence of underweight, overweight and obesity among children, early and late adolescents by school type using different criteria in total subjects.**

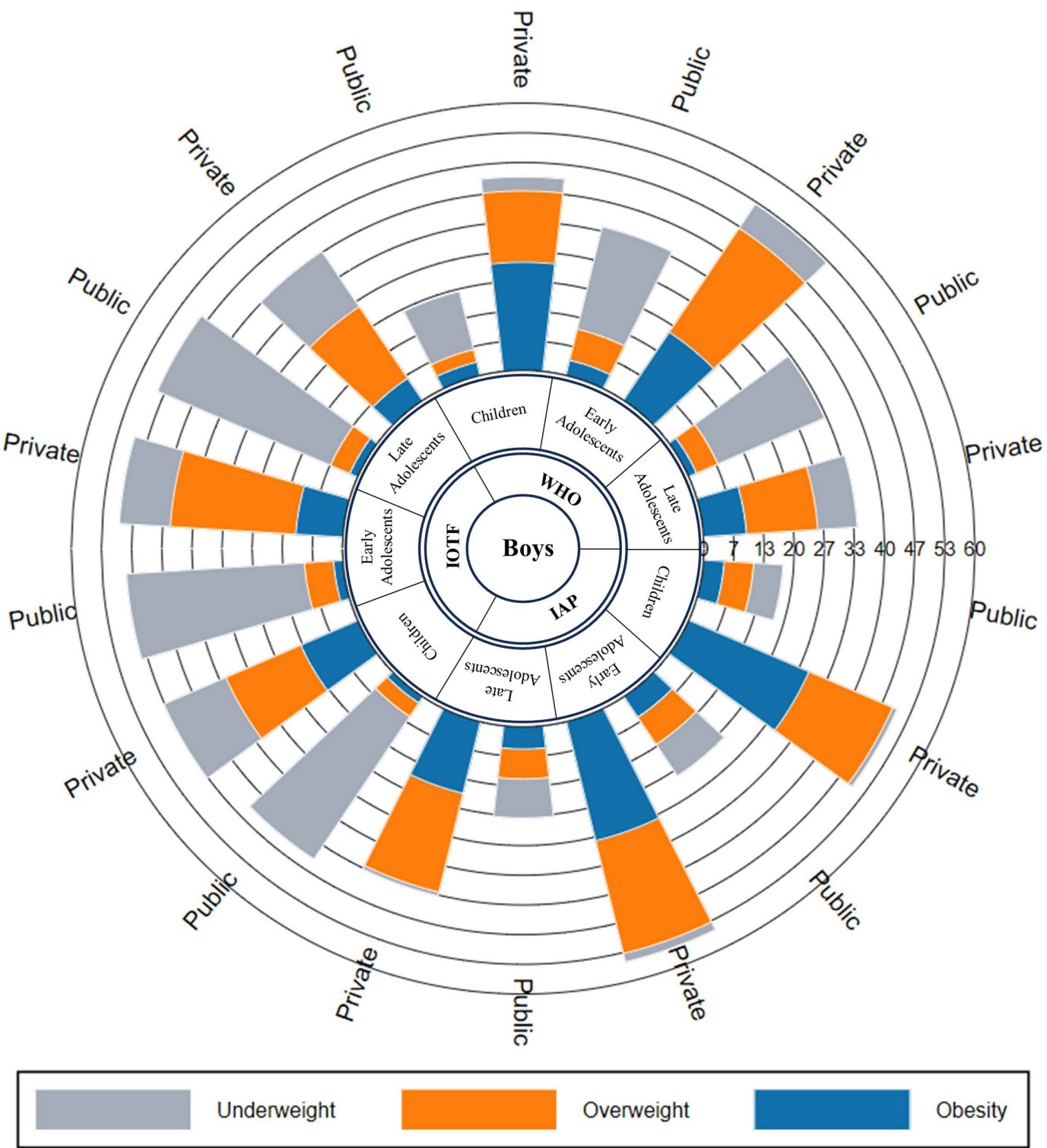

**Fig 3. Prevalence of underweight, overweight and obesity among children, early and late adolescents by school type using different criteria in boys.**

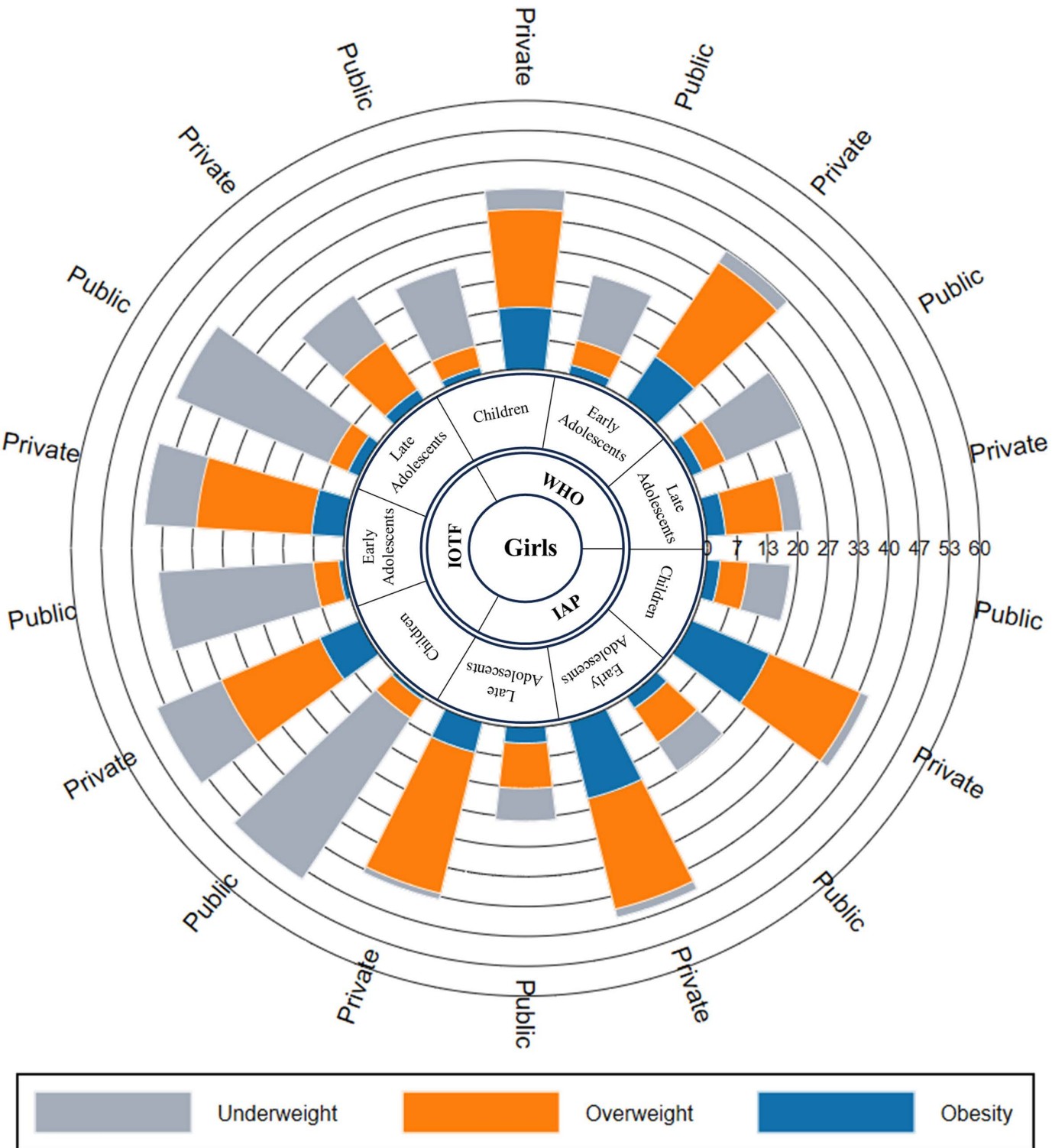

**Fig 4. Prevalence of underweight, overweight and obesity among children, early and late adolescents by school type using different criteria in girls.**

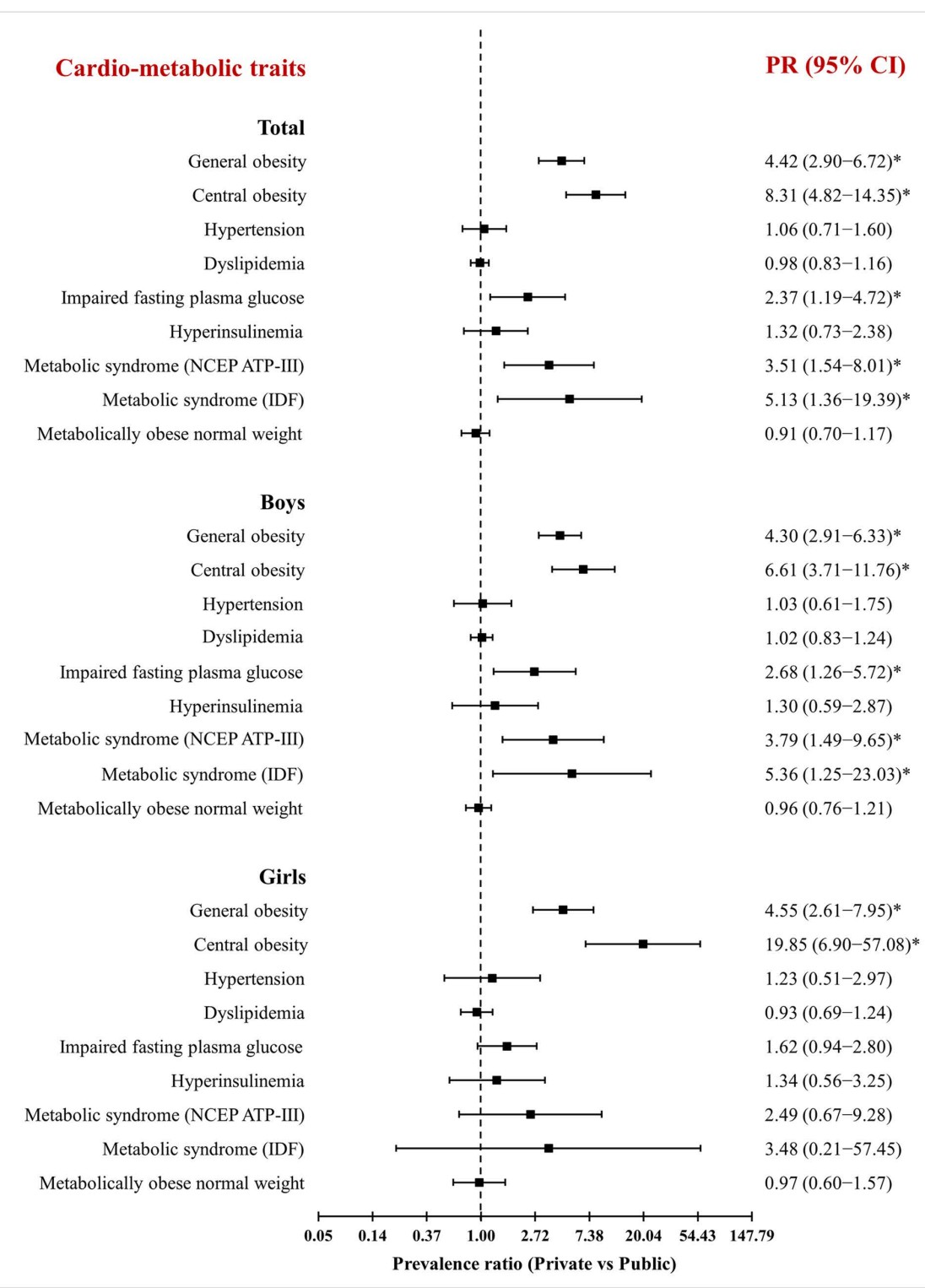

**Fig 5. Association between cardio-metabolic traits and the school type (reference category: public schools) among adolescents aged 10-19 years.** The association was assessed by random effect logistic regression analysis adjusted for children's age and gender and their father's and mother's education. The Prevalence Ratio (PR) of the cardio-metabolic traits was presented as the square, 95% confidence intervals (CI) by the lines through the squares, *p<0.05, statistically significant. General and central obesity were assessed among children and adolescents aged 6–19 years.

## Prevalence of lipid abnormality

The prevalence of low HDL-c (23.12%) among adolescents was high compared to the prevalence of other lipid levels (high TC = 4.41%, high TG = 7.97%, and high LDL-c = 2.91%). The prevalence of high TG was significantly higher (p = 0.030) among adolescents in private schools compared to public schools (Table 2). The difference was statistically significant among boys (p = 0.001) but not among girls (p = 0.525) (Tables 3 and 4).

## Prevalence of impaired fasting plasma glucose and hyperinsulinemia

The prevalence of impaired fasting plasma glucose (>100 mg/dl) and hyperinsulinemia (≥25 µIU/ml) among adolescents (Table 2) was 15.02% (95% CI 13.55–16.58) and 3.87% (95% CI 3.10–4.76) respectively. The prevalence of impaired fasting plasma glucose was more than twice as high in private school students compared to public schools with adjusted PR = 2.37 (95% CI 1.19–4.72). The prevalence of hyperinsulinemia was significantly lower in public school boys compared to private school (2.46% vs 4.91%, p = 0.018) boys (Table 3).

## Prevalence of dyslipidemia

The overall prevalence of dyslipidemia among adolescents was 33.73% (95% CI 31.75–35.75). The prevalence was significantly higher in public schools than in private schools (p = 0.036), with an adjusted PR of 0.98 (95% CI 0.83–1.16). Among girls, the prevalence of dyslipidemia was significantly higher in public than private schools (34.90% vs 26.61%, p = 0.008) (Table 4). The difference in the prevalence of high triglyceride was significant between public and private schools (7.03% vs 9.28%, p = 0.030) (Table 2). Low HDL-c was the most prevalent lipid abnormality (in both public and private school students) compared to the other lipid abnormalities, which might contribute to the high prevalence of dyslipidemia.

## Prevalence of MetS

The overall prevalence of MetS among urban adolescents according to NCEP ATP-III criteria (Table 2) was 3.43% (95% CI 2.70–4.28) and significantly lower among public school (1.75%) students compared to private school (5.76%) students according to the NCEP ATP-III (p < 0.001) with adjusted PR of 3.51 (95% CI 1.54–8.01) (Fig 5). A similar pattern (Tables 3 and 4) was observed between public and private schools among boys (p < 0.001) and girls (p = 0.010) using NCEP ATP-III criteria. The overall prevalence of MetS by the IDF criteria was 1.39% (95% CI 0.94–1.98), with public schools (0.40%) having significantly lower prevalence compared to private (2.77%) schools (p < 0.001) and the adjusted PR of 5.13 (95% CI 1.36–19.39) (Fig 5). Also, the prevalence was significantly lower in public schools compared to private schools in both boys (p < 0.001) and girls (p = 0.021).

## Metabolically obese normal weight/underweight phenotype

Among adolescents aged 10–19 years, 5.31% (115/2166) had a BMI < 5th percentile (underweight), and 67.54% (1463/2166) had a normal BMI according to IAP criteria. The mean levels of WC (p < 0.001), TC (p = 0.001), HDL-c (p < 0.001), LDL-c (p = 0.002), fasting plasma glucose (p < 0.001) and fasting insulin (p < 0.001) were significantly different among adolescents with MONW phenotype between private and public schools (Table 5). The prevalence of metabolically obese underweight (MOUW) among adolescents aged 10–19 years was 41.74% (95% CI 32.61–51.30), of which 89.58% (43/48) belong to public schools. The shift in the distribution of cardio-metabolic traits, except LDL-c, was statistically significant between MONW and MNNW (Metabolically Normal Normal weight) phenotypes among adolescents having normal BMI-for-age (Fig 6). The prevalence of MONW phenotype was 42.86% (95% CI 40.30–45.44), of which 73.68% (462/627) belong to public schools. There were no statistically significant differences in the prevalence of MONW for age (early vs late adolescents), sex, or school type. This phenotype was significantly more prevalent among public school

**Table 2. Prevalence of cardio-metabolic traits for adolescents aged 10-19 years in Delhi, by school type.**

| Cardio-metabolic traits | Prevalence (95% CI) [in %] | | | p-value |
|---|---|---|---|---|
| | *Total* | *Public* | *Private* | |
| **Physiological parameters@** | *n = 3856* | *n = 1966* | *n = 1890* | |
| *Body Mass Index (kg/m2)* | | | | |
| Underweight | 4.95 (4.29 − 5.69) | 8.09 (6.92 − 9.38) | 1.69 (1.16 − 2.38) | <0.001 |
| Normal | 65.98 (64.46 − 67.47) | 79.81 (77.96 − 81.56) | 51.59 (49.31 − 53.86) | |
| Overweight | 15.66 (14.53 − 16.85) | 7.63 (6.49 − 8.89) | 24.02 (22.11 − 26.01) | |
| Obesity | 13.41 (12.35 − 14.52) | 4.48 (3.61 − 5.49) | 22.70 (20.83 − 24.65) | |
| *Waist-circumference (cm)* | | | | |
| Normal | 90.85 (89.89 − 91.74) | 98.17 (97.47 − 98.71) | 83.23 (81.47 − 84.87) | <0.001 |
| Obesity | 9.15 (8.26 − 10.11) | 1.83 (1.29 − 2.53) | 16.77 (15.11 − 18.53) | |
| **Clinical Parameters** | *n = 2281* | *n = 1304* | *n = 977* | |
| *Blood Pressure (mmHg)* | | | | |
| Normal | 83.43 (81.84 − 84.93) | 83.90 (81.79 − 85.85) | 82.80 (80.29 − 85.12) | 0.586 |
| Pre-hypertensive | 9.21 (8.05 − 10.47) | 8.67 (7.19 − 10.33) | 9.93 (8.13 − 11.98) | |
| Hypertensive | 7.37 (6.33 − 8.51) | 7.44 (6.07 − 9.00) | 7.27 (5.72 − 9.08) | |
| **Biochemical parameters** | *n = 2197* | *n = 1281* | *n = 916* | |
| *Total cholesterol (mg/dl)* | | | | |
| Normal | 80.47 (78.75 − 82.11) | 85.79 (83.76 − 87.66) | 73.03 (70.04 − 75.89) | <0.001 |
| Borderline | 15.11 (13.64 − 16.67) | 10.85 (9.20 − 12.68) | 21.07 (18.47 − 23.86) | |
| High | 4.41 (3.59 − 5.36) | 3.36 (2.44 − 4.50) | 5.90 (4.46 − 7.62) | |
| *Triglyceride (mg/dl)* | | | | |
| Normal | 68.18 (66.19 − 70.13) | 70.26 (67.67 − 72.75) | 65.28 (62.10 − 68.37) | 0.030 |
| Borderline | 23.85 (22.08 − 25.69) | 22.72 (20.45 − 25.11) | 25.44 (22.64 − 28.39) | |
| High | 7.97 (6.87 − 9.18) | 7.03 (5.69 − 8.57) | 9.28 (7.48 − 11.35) | |
| *Low-density lipoprotein (mg/dl)* | | | | |
| Normal | 94.81 (93.80 − 95.70) | 92.74 (91.18 − 94.10) | 97.71 (96.52 − 98.58) | <0.001 |
| Borderline | 2.28 (1.69 − 2.99) | 2.89 (2.04 − 3.96) | 1.42 (0.76 − 2.41) | |
| High | 2.91 (2.25 − 3.70) | 4.37 (3.32 − 5.64) | 0.87 (0.38 − 1.71) | |
| *High-density lipoprotein (mg/dl)* | | | | |
| Normal | 53.76 (51.64 − 55.86) | 50.27 (47.50 − 53.05) | 58.62 (55.36 − 61.84) | <0.001 |
| Borderline | 23.12 (21.37 − 24.94) | 24.2 (21.88 − 26.64) | 21.62 (18.99 − 24.43) | |
| Low | 23.12 (21.37 − 24.94) | 25.53 (23.16 − 28.01) | 19.76 (17.23 − 22.49) | |
| *Fasting plasma glucose (mg/dl)* | | | | |
| Normal | 84.98 (83.42 − 86.45) | 89.07 (87.23 − 90.73) | 79.26 (76.49 − 81.84) | <0.001 |
| Impaired | 15.02 (13.55 − 16.58) | 10.93 (9.27 − 12.77) | 20.74 (18.16 − 23.51) | |
| *Fasting insulin (µIU/ml)* | | | | |
| Normal | 96.13 (95.24 − 96.90) | 97.11 (96.04 − 97.96) | 94.76 (93.11 − 96.11) | 0.005 |
| High | 3.87 (3.10 − 4.76) | 2.89 (2.04 − 3.96) | 5.24 (3.89 − 6.89) | |
| **Dyslipidemia** | 33.73 (31.75 − 35.75) | 35.52 (32.9 − 38.21) | 31.22 (28.23 − 34.34) | 0.036 |
| **MetS#** | *n = 2160* | *n = 1257* | *n = 903* | |
| *NCEP ATP-III* | 3.43 (2.70 − 4.28) | 1.75 (1.10 − 2.64) | 5.76 (4.33 − 7.48) | <0.001 |
| *IDF* | 1.39 (0.94 − 1.98) | 0.40 (0.13 − 0.93) | 2.77 (1.80 − 4.06) | <0.001 |
| **MONW\*** | *n = 1463* | *n = 996* | *n = 467* | |
| *At-least one$* | 42.86 (40.30 − 45.44) | 46.39 (43.25 − 49.54) | 35.33 (30.99 − 39.86) | <0.001 |

@Physiological parameters for children aged 6–19 years,

#Metabolic syndrome,

\*Metabolically obese normal weight:

$Abnormality in any of the following: Waist Circumference, Blood pressure, Triglyceride, High-density lipoprotein-cholesterol (mg/dl), and Fasting plasma glucose, NCEP ATP-III: National Cholesterol Education Program (NCEP) Adult Treatment Panel-III, IDF: International Diabetes Federation

**Table 3. Prevalence of cardio-metabolic traits for boys aged 10-19 years in Delhi, by school type.**

| Cardio-metabolic traits | Prevalence (95% CI) [in %] | | | p-value |
|---|---|---|---|---|
| | Total | Public | Private | |
| **Physiological parameters@** | **n = 2215** | **n = 1152** | **n = 1063** | |
| *Body Mass Index (kg/m2)* | | | | |
| Underweight | 4.92 (4.06 – 5.91) | 8.07 (6.57 – 9.80) | 1.51 (0.86 – 2.43) | <0.001 |
| Normal | 64.33 (62.30 – 66.33) | 79.51 (77.07 – 81.81) | 47.88 (44.84 – 50.94) | |
| Overweight | 14.99 (13.53 – 16.54) | 7.03 (5.62 – 8.66) | 23.61 (21.09 – 26.28) | |
| Obesity | 15.76 (14.26 – 17.34) | 5.38 (4.15 – 6.85) | 27.00 (24.35 – 29.78) | |
| *Waist-circumference (cm)* | | | | |
| Normal | 88.71 (87.32 – 90.00) | 97.31 (96.20 – 98.16) | 79.40 (76.84 – 81.79) | <0.001 |
| Obesity | 11.29 (10.00 – 12.68) | 2.69 (1.84 – 3.80) | 20.60 (18.21 – 23.16) | |
| **Clinical Parameters** | **n = 1355** | **n = 783** | **n = 572** | |
| *Blood Pressure (mmHg)* | | | | |
| Normal | 85.24 (83.24 – 87.09) | 85.70 (83.05 – 88.07) | 84.62 (81.39 – 87.47) | 0.842 |
| Pre-hypertensive | 7.60 (6.25 – 9.14) | 7.28 (5.56 – 9.33) | 8.04 (5.95 – 10.58) | |
| Hypertensive | 7.16 (5.84 – 8.66) | 7.02 (5.34 – 9.05) | 7.34 (5.34 – 9.80) | |
| **Biochemical parameters** | **n = 1300** | **n = 771** | **n = 529** | |
| *Total cholesterol (mg/dl)* | | | | |
| Normal | 83.08 (80.93 – 85.08) | 88.98 (86.55 – 91.10) | 74.48 (70.54 – 78.14) | <0.001 |
| Borderline | 12.77 (11.00 – 14.71) | 8.69 (6.80 – 10.90) | 18.71 (15.48 – 22.30) | |
| High | 4.15 (3.14 – 5.39) | 2.33 (1.39 – 3.66) | 6.81 (4.81 – 9.30) | |
| *Triglyceride (mg/dl)* | | | | |
| Normal | 70.54 (67.98 – 73.01) | 74.32 (71.08 – 77.37) | 65.03 (60.79 – 69.09) | 0.001 |
| Borderline | 21.69 (19.48 – 24.03) | 19.07 (16.35 – 22.02) | 25.52 (21.86 – 29.46) | |
| High | 7.77 (6.37 – 9.36) | 6.61 (4.96 – 8.61) | 9.45 (7.10 – 12.27) | |
| *Low-density lipoprotein (mg/dl)* | | | | |
| Normal | 93.54 (92.06 – 94.81) | 90.53 (88.24 – 92.50) | 97.92 (96.31 – 98.96) | <0.001 |
| Borderline | 2.62 (1.82 – 3.64) | 3.50 (2.32 – 5.05) | 1.32 (0.53 – 2.71) | |
| High | 3.85 (2.87 – 5.04) | 5.97 (4.40 – 7.88) | 0.76 (0.21 – 1.92) | |
| *High-density lipoprotein (mg/dl)* | | | | |
| Normal | 52.08 (49.32 – 54.82) | 49.42 (45.83 – 53.01) | 55.95 (51.61 – 60.24) | 0.065 |
| Borderline | 22.85 (20.59 – 25.23) | 23.87 (20.90 – 27.04) | 21.36 (17.94 – 25.10) | |
| Low | 25.08 (22.74 – 27.53) | 26.72 (23.62 – 29.99) | 22.68 (19.18 – 26.49) | |
| *Fasting plasma glucose (mg/dl)* | | | | |
| Normal | 83.85 (81.73 – 85.81) | 88.59 (86.13 – 90.74) | 76.94 (73.11 – 80.46) | <0.001 |
| Impaired | 16.15 (14.19 – 18.27) | 11.41 (9.26 – 13.87) | 23.06 (19.54 – 26.89) | |
| *Fasting insulin (µIU/ml)* | | | | |
| Normal | 96.54 (95.40 – 97.46) | 97.54 (96.18 – 98.51) | 95.09 (92.88 – 96.76) | 0.018 |
| High | 3.46 (2.54 – 4.60) | 2.46 (1.49 – 3.82) | 4.91 (3.24 – 7.12) | |
| **Dyslipidemia** | 35.38 (32.78 – 38.05) | 35.93 (32.54 – 39.43) | 34.59 (30.54 – 38.82) | 0.621 |
| **MetS#** | **n = 1276** | **n = 752** | **n = 524** | |
| NCEP ATP-III | 3.84 (2.85 – 5.05) | 1.86 (1.02 – 3.10) | 6.68 (4.70 – 9.17) | <0.001 |
| IDF | 1.80 (1.15 – 2.69) | 0.53 (0.15 – 1.36) | 3.63 (2.20 – 5.60) | <0.001 |
| **MONW*** | **n = 850** | **n = 594** | **n = 256** | |
| At-least one$ | 41.65 (38.31 – 45.04) | 44.61 (40.57 – 48.71) | 34.77 (28.94 – 40.95) | 0.008 |

@Physiological parameters for boys aged 6–19 years,

#Metabolic syndrome,

*Metabolically obese normal weight:

$Abnormality in any of the following: Waist Circumference, Blood pressure, Triglyceride, High-density lipoprotein-cholesterol (mg/dl), and Fasting plasma glucose, NCEP ATP-III: National Cholesterol Education Program (NCEP) Adult Treatment Panel-III, IDF: International Diabetes Federation

**Table 4. Prevalence of cardio-metabolic traits for girls aged 10-19 years in Delhi, by school type.**

| Cardio-metabolic traits | Prevalence (95% CI) [in %] | | | p-value |
|---|---|---|---|---|
| | Total | Public | Private | |
| Physiological parameters@ | n=1641 | n=814 | n=827 | |
| Body Mass Index (kg/m²) | | | | |
| Underweight | 5.00 (3.99−6.16) | 8.11 (6.33−10.20) | 1.93 (1.11−3.12) | <0.001 |
| Normal | 68.19 (65.88−70.44) | 80.22 (77.32−82.91) | 56.35 (52.89−59.76) | |
| Overweight | 16.58 (14.81−18.46) | 8.48 (6.66−10.61) | 24.55 (21.65−27.63) | |
| Obesity | 10.24 (8.81−11.81) | 3.19 (2.10−4.65) | 17.17 (14.66−19.92) | |
| Waist-circumference (cm) | | | | |
| Normal | 93.72 (92.44−94.85) | 99.39 (98.57−99.8) | 88.15 (85.75−90.27) | <0.001 |
| Obesity | 6.28 (5.15−7.56) | 0.61 (0.20−1.43) | 11.85 (9.73−14.25) | |
| Clinical Parameters | n=926 | n=521 | n=405 | |
| Blood Pressure (mmHg) | | | | |
| Normal | 80.78 (78.09−83.27) | 81.19 (77.56−84.46) | 80.25 (76.03−84.01) | 0.626 |
| Pre-hypertensive | 11.56 (9.57−13.79) | 10.75 (8.22−13.73) | 12.59 (9.52−16.22) | |
| Hypertensive | 7.67 (6.04−9.57) | 8.06 (5.87−10.74) | 7.16 (4.85−10.12) | |
| Biochemical parameters | n=897 | n=510 | n=387 | |
| Total cholesterol (mg/dl) | | | | |
| Normal | 76.70 (73.79−79.43) | 80.98 (77.30−84.30) | 71.06 (66.26−75.53) | 0.001 |
| Borderline | 18.51 (16.02−21.21) | 14.12 (11.21−17.45) | 24.29 (20.10−28.88) | |
| High | 4.79 (3.49−6.40) | 4.90 (3.20−7.15) | 4.65 (2.78−7.25) | |
| Triglyceride (mg/dl) | | | | |
| Normal | 64.77 (61.55−67.90) | 64.12 (59.78−68.29) | 65.63 (60.67−70.36) | 0.525 |
| Borderline | 26.98 (24.10−30.01) | 28.24 (24.37−32.36) | 25.32 (21.06−29.96) | |
| High | 8.25 (6.53−10.25) | 7.65 (5.49−10.31) | 9.04 (6.38−12.35) | |
| Low-density lipoprotein (mg/dl) | | | | |
| Normal | 96.66 (95.26−97.73) | 96.08 (94.01−97.59) | 97.42 (95.30−98.75) | 0.482 |
| Borderline | 1.78 (1.02−2.88) | 1.96 (0.94−3.58) | 1.55 (0.57−3.34) | |
| High | 1.56 (0.86−2.60) | 1.96 (0.94−3.58) | 1.03 (0.28−2.63) | |
| High-density lipoprotein (mg/dl) | | | | |
| Normal | 56.19 (52.87−59.47) | 51.57 (47.13−55.98) | 62.27 (57.24−67.12) | 0.002 |
| Borderline | 23.52 (20.78−26.44) | 24.71 (21.02−28.69) | 21.96 (17.94−26.42) | |
| Low | 20.29 (17.70−23.07) | 23.73 (20.10−27.66) | 15.76 (12.28−19.78) | |
| Fasting plasma glucose (mg/dl) | | | | |
| Normal | 86.62 (84.22−88.78) | 89.80 (86.84−92.29) | 82.43 (78.26−86.09) | 0.001 |
| Impaired | 13.38 (11.22−15.78) | 10.20 (7.71−13.16) | 17.57 (13.91−21.74) | |
| Fasting insulin (µIU/ml) | | | | |
| Normal | 95.54 (93.98−96.80) | 96.47 (94.48−97.90) | 94.32 (91.52−96.40) | 0.121 |
| High | 4.46 (3.20−6.02) | 3.53 (2.10−5.52) | 5.68 (3.60−8.48) | |
| Dyslipidemia | 31.33 (28.30−34.48) | 34.90 (30.76−39.22) | 26.61 (22.28−31.32) | 0.008 |
| MetS# | n=884 | n=505 | n=379 | |
| NCEP ATP-III | 2.83 (1.84−4.15) | 1.58 (0.69−3.10) | 4.49 (2.63−7.08) | 0.010 |
| IDF | 0.79 (0.32−1.62) | 0.20 (0.01−1.10) | 1.58 (0.58−3.41) | 0.021 |
| MONW* | n=613 | n=402 | n=211 | |
| At-least one$ | 44.53 (40.55−48.57) | 49.00 (44.02−54.01) | 36.02 (29.54−42.89) | 0.002 |

@Physiological parameters for girls aged 6-19 years, #Metabolic syndrome, *Metabolically obese normal weight: $Abnormality in any of the following: Waist Circumference, Blood pressure, Triglyceride, High-density lipoprotein-cholesterol (mg/dl), and Fasting plasma glucose, NCEP ATP-III: National Cholesterol Education Program (NCEP) Adult Treatment Panel-III, IDF: International Diabetes Federation

**Table 5.** Comparison of cardio-metabolic traits among adolescents with MONW phenotype between public and private schools.

| Cardio-metabolic traits | Adolescents (10-19 years) with MONW phenotype | | | | p-value |
|---|---|---|---|---|---|
| | Total mean±SD | Public mean±SD | Private mean±SD | Difference (95% CI) | |
| | n=627 | n=462 | n=165 | | |
| Waist Circumference (cm) | 62.63±6.62 | 61.86±6.79 | 64.80±5.62 | 2.94 (1.78−4.10) | <0.001 |
| Systolic blood pressure (mmHg) | 106.15±10.03 | 105.74±10.10 | 107.27±9.77 | 1.53 (-0.26−3.31) | 0.093 |
| Diastolic blood pressure (mmHg) | 69.22±8.17 | 68.91±8.10 | 70.09±8.33 | 1.18 (-0.27−2.64) | 0.111 |
| Total cholesterol (mg/dl) | 138.60±29.99 | 136.13±28.96 | 145.49±31.78 | 9.36 (4.06−14.65) | 0.001 |
| High-density lipoprotein (mg/dl) | 41.85±9.86 | 40.97±8.77 | 44.32±12.11 | 3.34 (1.60−5.08) | <0.001 |
| TC to HDL-c ratio | 3.38±0.67 | 3.38±0.65 | 3.39±0.72 | 0.01 (-0.11−0.13) | 0.861 |
| Triglyceride (mg/dl) | 89.21±37.11 | 90.03±37.59 | 86.93±35.74 | -3.09 (-9.70−3.52) | 0.359 |
| Low-density lipoprotein (mg/dl) | 78.90±23.98 | 77.16±23.75 | 83.79±24.02 | 6.63 (2.39−10.87) | 0.002 |
| Fasting plasma glucose (mg/dl) | 92.93±9.45 | 91.48±8.07 | 96.96±11.64 | 5.48 (3.85−7.11) | <0.001 |
| Fasting insulin (µIU/ml) | 10.12±7.46 | 9.44±5.59 | 12.02±10.94 | 2.58 (1.26−3.89) | <0.001 |

TC: Total cholesterol, HDL-c: High-density lipoprotein-c

students 46.39% (95% CI 43.25–49.54) than private school students 35.33% (95% CI 30.99–39.86), with p < 0.001 having adjusted PR of 0.91 (95% CI 0.70–1.17) (Table 2 & Fig 5). The most prevalent cardio-metabolic abnormality among MONW phenotype between public and private schools was low HDL-c (62.12% vs 52.73, p = 0.039), followed by hypertension (29.28% vs 34.55%, p = 0.140), high TG (28.14% vs 25.45%, p = 0.523), high FPG (3.03% vs 5.45%, p = 0.377) and hyperinsulinemia (1.95% vs 2.42%, p = 0.582) adolescents aged 10–19 years (Fig 7).

## Discussion

To the best of our knowledge, this is the first longitudinal school-based study that comprehensively assessed the cardio-metabolic traits such as anthropometry, blood pressure, lipids, fasting plasma glucose and fasting insulin among private and public schools in India using robust and standard methodologies for blood sample analysis and data collection while addressing the limitations of previous studies. The findings revealed that the prevalence of being underweight was nearly five times higher in public school compared to private school students. At the same time, obesity was more than five times higher in private school compared to public school students. Obesity prevalence was higher while using IAP criteria (developed using data from Indian children) compared to the WHO and IOTF criteria that are based on the data from the Caucasian population. Private school students (compared to public school students) had a higher risk for general obesity, central obesity, impaired fasting plasma glucose and MetS after adjusting for children's age and gender and their father's and mother's education, suggesting that they were more vulnerable for cardio-metabolic risk. On the contrary, among adolescents aged 10–19 years having normal and underweight BMI-for-age, over two-fifths of students had at least one metabolic biomarker abnormal, with a higher prevalence observed among public school students than private school students. The most prevalent biomarker was low HDL-c, which was noticeably more prevalent among public school students.

The strength of this study lies in its comprehensive assessment of cardio-metabolic traits and estimation of the prevalence of childhood underweight and overweight/obesity, notably underweight prevalence using IAP criteria, along with the WHO and IOTF criteria among children and adolescents aged 6–19 years from both private and public schools. To the best of our knowledge, this is the first study in the Indian context to report socioeconomic differential in the prevalence of the MONW phenotype among normal-weight adolescents (BMI-for-age, IAP criteria) aged 10–19 years. Also, this is the first study conducted in the post-COVID era providing prevalence estimates of various cardio-metabolic traits

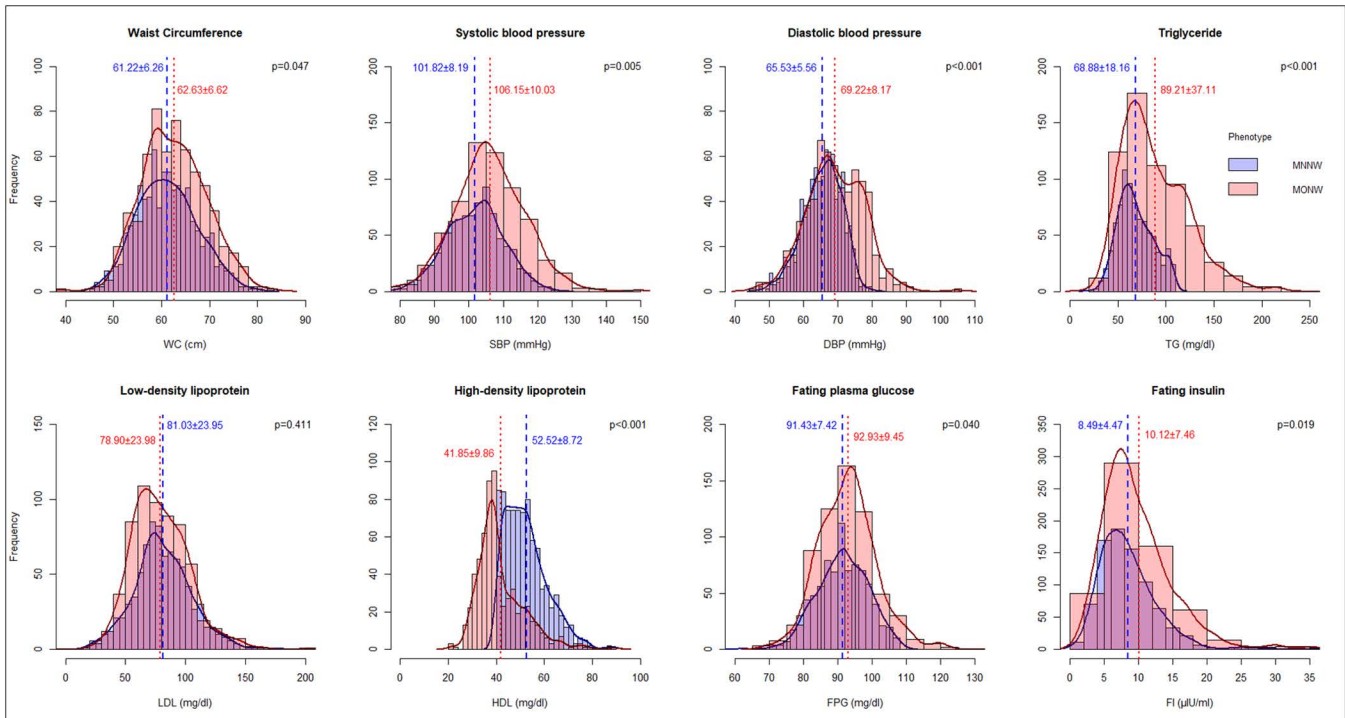

**Fig 6. Comparative distribution of cardio-metabolic traits among adolescents aged 10-19 years by MNNW (n = 836) and MONW (627) phenotypes.**

among children and adolescents aged 6–19 years in Delhi, the capital and a metropolitan city that represents the socio-cultural diversity of India. These findings are crucial for understanding the adverse effects of lifestyle changes following the COVID-19 lockdown on nutrition, especially among vulnerable and marginalized groups. Although the current study lacks pre-pandemic data to attribute these effects directly, evidence suggests that adolescents experienced increased BMI due to unhealthy eating behaviors, disrupted sleep patterns, reduced physical activity, and increased screen time during the prolonged COVID-19 lockdown [39]. As the sample size was not determined to estimate the prevalence of cardio-metabolic traits by socioeconomic status, the confidence intervals for some prevalence estimates appear to be wide. Selection bias is likely due to the non-random selection of schools, which is the limitation of the study. Moreover, the high non-response among affluent children and adolescents may have impacted prevalence estimates, despite multiple reminders to those students' parents by email/WhatsApp class group and a session by a study representative with the parents during parents-teacher meetings, primarily due to parental hesitance about medical procedures like blood sampling in a school setting. This reluctance, particularly among parents from higher socioeconomic backgrounds, was heightened by the unprecedented circumstances of the COVID-19 pandemic.

Our study contributes to fill the gap of scarce data on underweight for post-COVID-19 period [20]. In this study, the prevalence of underweight using WHO criteria was slightly lower (12.24%) than in a similar study from Delhi (13.03%) [31]. Among late adolescents, underweight prevalence (author-defined IAP criteria<5th percentile) was 5.53%, which is higher than the 3.90% reported in the previous study [40], possibly due to the use of the 5th percentile threshold. However, IAP does not specify underweight criteria. National data from Comprehensive National Nutrition Survey (CNNS) (2016–18) reported higher rates (24.10% in India and 21.30% in Delhi using the WHO criteria) among adolescents aged 10–19 years [41]. Noticeably, the current study reported a lower prevalence of underweight among public school children

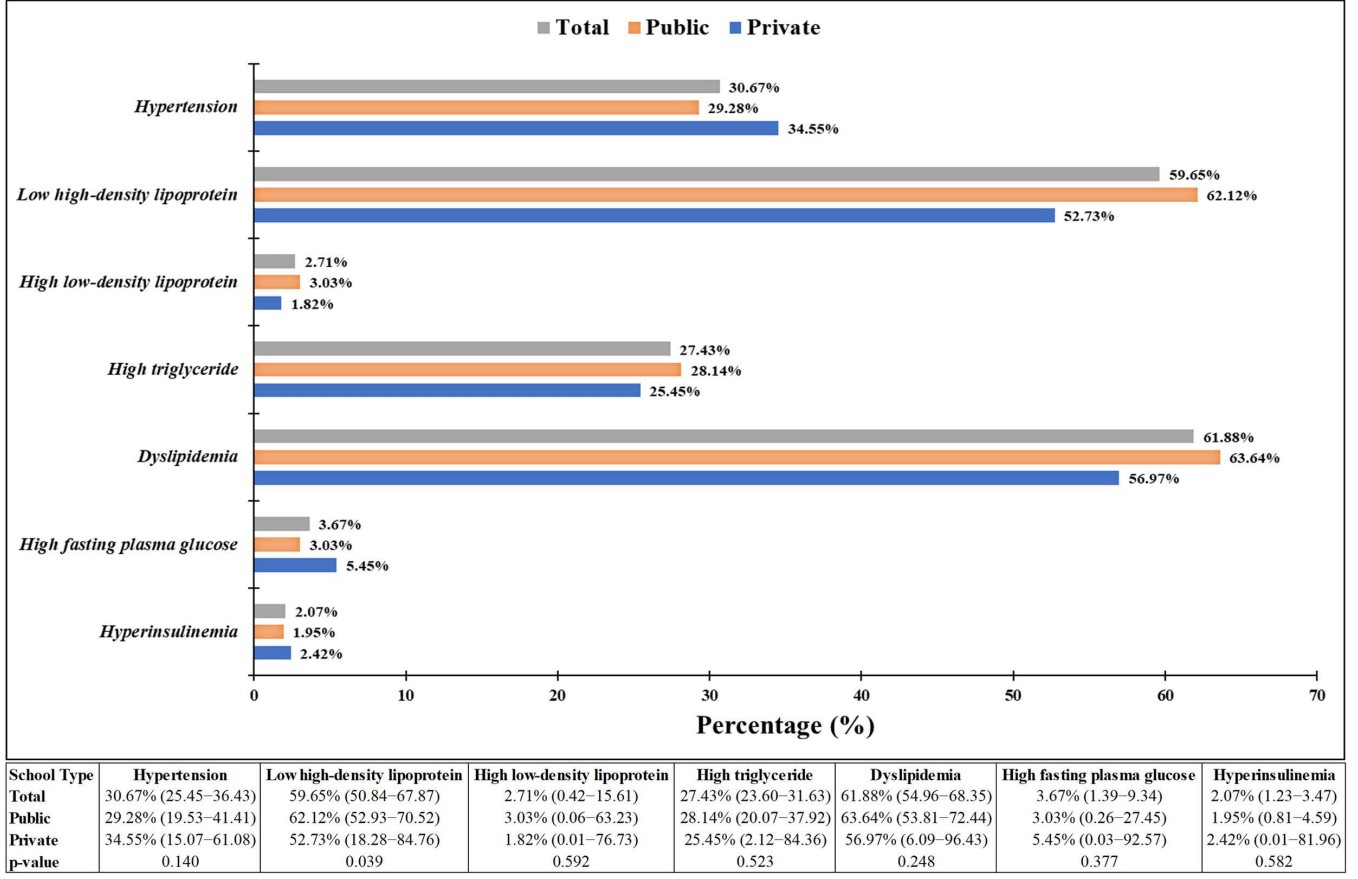

**Fig 7. Comparison of cardio-metabolic traits by school type among metabolically obese normal weight phenotype in adolescents aged 10-19 years (n = 1463).**

compared to the varied underweight prevalence reported in previous studies [31,42,43]. Similarly, the current study reported lower rates of underweight prevalence among private school children than reported previously [31,44–47]. This suggests a declining trend in underweight prevalence among public school children, with differences between public and private schools likely attributed to highly interlinked socioeconomic, lifestyle and nutritional disparities.

Globally, childhood obesity was found to be directly proportional to SES [48], and consistent with this, the present study findings showed a similar relationship with both general obesity [31,40,49–53] and central obesity [31,54]. The prevalence of overweight, obesity, and combined overweight and obesity reported from the present study (15.66%, 13.41%, and 29.07%, respectively) were higher than the pooled prevalence estimated for India using the IAP criteria (15.20%, 9.20%, and 25.20%, respectively) [55]. Also, compared to a limited number of studies from India that had reported the prevalence of central obesity among children and adolescents, the present study reported a higher prevalence of central obesity among both private school and public school students [31,54,56–59]. Evidently, all these studies were conducted before the COVID-19 pandemic except the current study. Thus, the increase in prevalence may be attributed to the adverse effects of the COVID-19 pandemic, which fostered a more sedentary lifestyle for children and adolescents along with other known factors such as rapid urbanization, changing patterns of diet and reduced physical activity [40,54,60,61].

Though dyslipidemia is a key component of cardio-metabolic risk, in India, research on lipid abnormalities among adolescents is limited and often focuses on high-risk groups. The findings of this study align with previous research

[31,62–64], showing overall higher rates of low HDL-c and high LDL-c, along with lower rates of high TG among adolescents from lower SES backgrounds. These differences may be attributed to food scarcity, leading to poor dietary choices as these families prioritize calorie intake over nutritional quality, highlighting the critical need for food security. Specifically, boys exhibited a higher prevalence of high LDL-c and lower rates of high TG. At the same time, girls had higher rates of low HDL-c, suggesting distinct pathogenic mechanisms within the atherogenic triad. The higher prevalence of low HDL-c among girls belonging to lower SES may be linked to cortisol excretion; thus, a disproportionate stress response among females, coupled with the higher prevalence of depression-which is also associated with low HDL-c could contribute to a significant lowering of HDL-c levels [65–68]. Given this complexity, further research is needed to understand better the relationship between stress, dietary patterns, gender, and lipid levels and explore these interactions in greater depth among adolescents.

The present study reported an overall prevalence of 3.43% for MetS, which is lower than the pooled prevalence estimated for India and the global level [69–71]. The prevalence of MetS among private school students (affluent) was higher (5.76%) than among public school students (1.75%) and was also higher than the estimates for both India and the global level [70,71]. This finding compares with population-based data from India where adolescents from the affluent strata had higher odds of MetS than their counterparts. This could be due to the high prevalence of obesity among affluent adolescents linked with their sedentary behavior and easy access to junk foods and aerated drinks [70]. Also consistent with previous studies, boys (3.84%) had a higher prevalence of MetS than girls (2.83%) [69]. This could be due to sex differences in dietary intake and appetite traits arising from the differential influence of biopsychosocial factors favoring unhealthy behaviors for boys [72]. The findings highlight the need for active surveillance for MetS among adolescents along with gender-sensitive and targeted lifestyle interventions to mitigate the long-term impact of cardio-metabolic risks on adult health.

Given the preponderance of MetS among overweight or obese individuals, its occurrence in those with normal weight was often overlooked until 1981 [30], when the MONW (Metabolically Obese Normal Weight) phenotype was defined with the presence of at least one metabolic abnormality among normal BMI-for-age individuals. Though there are few global studies among children and adolescents on MONW phenotype [73–75], it is relatively unexplored in India [76,77]. The current study found that more than two-fifths of normal weight (42.86%) and underweight (41.74%) adolescents had metabolic obesity, a rate lower than those reported in a previous study in India [76,77]. This variation could be attributed to differences in the parameters included or the cut-off points used to define metabolic obesity. The current study defined MONW as the presence of at least one metabolic abnormality defined for MetS among normal BMI-for-age individuals, which is in line with the conventional definition of MetS. The prevalence of MONW reported in the current study highlights the limitation of conventional anthropometric parameters in assessing the cardio-metabolic risk among adolescents, and future data from LEAP-C, which measures various anthropometric parameters longitudinally, may provide valuable insights on indices sensitive to cardio-metabolic risk among the adolescents. On further analysis, higher rates of MONW phenotype were reported among adolescents belonging to public schools (lower socioeconomic status) than their counterparts. Additionally, a differential pattern with a preponderance of glycemic-related abnormality among private school students and lipid-related abnormality (dyslipidemia) among public school students was noted. This could be attributed to food insecurity among adolescents from lower socioeconomic status, leading to increased consumption of added sugar and refined carbohydrates, and warrants further research to characterize potential interactions between dietary intake and dyslipidemia in adolescents [67].

Besides the above findings, longitudinal data from LEAP-C will explore the correlation between anthropometric measurements and biochemical parameters to better understand CVD risk among adolescents within the Indian context. Next, using data collected from private school children, this study will provide longitudinal growth reference for anthropometric parameters, blood pressures, lipids, fasting plasma glucose and fasting insulin. Also, this study will enable us to establish the association in explaining the determinants of clustering of CVD risks, which cannot be achieved with the cross-sectional nature of baseline data.

In India, Safe Food and Balanced Diets for Children in School under Food Safety and Standards Regulations, 2020 aims to promote a balanced diet and reduce the availability of food high in fat, salt and sugar among school children [78], which is encouraging the initial step towards food security. India's child and adolescent health programs, such as Rashtriya Bal Swasthya Karyakram and Rashtriya Kishor Swasthya Karyakram [79], aims to promote health and well-being of young age groups. However, there is a need for a comprehensive approach to integrated interventions connecting multi-stakeholders such as schools, parents, communities and the healthcare system for effective implementation to address the growing burden of cardio-metabolic risks among children and adolescents. The observed higher rates of general and central obesity among boys and distinct lipid patterns between genders emphasize the need for gender-sensitive approaches in tackling cardio-metabolic risks. Ensuring food security and active surveillance of metabolic risks are thus critical to addressing the complex challenges of cardio-metabolic burden in young age groups.

## Conclusion

This study is a comprehensive effort to date in understanding the cardio-metabolic perturbations among adolescents from private and public schools in India. Obesity, a precursor of various NCDs, is notably prevalent among affluent adolescents, exceptionally high among younger children compared to adolescents and boys compared to girls. Substantially, high metabolic obesity among normal-weight and underweight children, particularly from public schools warrants early intervention. Therefore, future research efforts should focus on developing robust and sensitive parameters for early identification and active surveillance. Existing public health programs should be strengthened with focused approaches, such as ensuring food security, promoting physical activity and implementing periodic metabolic risk screening to reduce cardio-metabolic burden.

## Supporting Information

**Supporting Information. This file contains the supplementary tables and S1, S2 files.**
(DOCX)

## Acknowledgments

We gratefully acknowledge the Director (Education), New Delhi Municipal Council, school principals, and teachers for granting permission and supporting the conduct of the study. We thank all the field data collection and laboratory investigation teams. We gratefully acknowledge Senthil Amudhan, National Institute of Mental Health Sciences, Bengaluru, India. We also thank all the study participants and their parents for their support and cooperation in completing this study. We also thank the REDCap administrators for providing a user account and its management for storing the study data.

## Author contributions

**Conceptualization:** Mani Kalaivani, Raman Kumar Marwaha, Nikhil Tandon.

**Data curation:** Mani Kalaivani, Chitralok Hemraj.

**Formal analysis:** Mani Kalaivani, Chitralok Hemraj.

**Funding acquisition:** Mani Kalaivani, Nikhil Tandon.

**Investigation:** Mani Kalaivani, Chitralok Hemraj, Varhlunchhungi Varhlunchhungi.

**Methodology:** Mani Kalaivani, Chitralok Hemraj, Lakshmy Ramakrishnan, Sumit Malhotra, Sanjeev Kumar Gupta, Ransi Ann Abraham, Nikhil Tandon.

**Project administration:** Mani Kalaivani, Chitralok Hemraj, Varhlunchhungi Varhlunchhungi.

**Resources:** Mani Kalaivani, Lakshmy Ramakrishnan, Monika Arora, Tina Rawal, Maroof Ahmad Khan, Nikhil Tandon.

**Software:** Mani Kalaivani.

**Supervision:** Mani Kalaivani, Lakshmy Ramakrishnan, Ransi Ann Abraham.

**Validation:** Mani Kalaivani, Chitralok Hemraj, Varhlunchhungi Varhlunchhungi, Lakshmy Ramakrishnan, Sumit Malhotra, Sanjeev Kumar Gupta, Raman Kumar Marwaha, Ransi Ann Abraham, Monika Arora, Tina Rawal, Maroof Ahmad Khan, Aditi Sinha, Nikhil Tandon.

**Visualization:** Mani Kalaivani, Lakshmy Ramakrishnan, Sanjeev Kumar Gupta, Raman Kumar Marwaha, Nikhil Tandon.

**Writing – original draft:** Mani Kalaivani, Varhlunchhungi Varhlunchhungi.

**Writing – review & editing:** Mani Kalaivani, Chitralok Hemraj, Varhlunchhungi Varhlunchhungi, Lakshmy Ramakrishnan, Sumit Malhotra, Sanjeev Kumar Gupta, Raman Kumar Marwaha, Ransi Ann Abraham, Monika Arora, Tina Rawal, Maroof Ahmad Khan, Aditi Sinha, Nikhil Tandon.

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
