## [Decision Letter · Decision Letter 0]

26 Jan 2025

PONE-D-24-53240Cardio-metabolic traits and its socioeconomic differentials among school children including metabolically obese normal weight phenotypes in India: A post-COVID baseline characteristics of LEAP-C cohortPLOS ONE

Dear Dr. Mani,

Thank you for submitting your manuscript to PLOS ONE. After careful consideration, we feel that it has merit but does not fully meet PLOS ONE’s publication criteria as it currently stands. Therefore, we invite you to submit a revised version of the manuscript that addresses the points raised during the review process.

**ACADEMIC EDITOR: Dear Author, please make necessary changes based on the comments provided by the reviewer/s.**

We look forward to receiving your revised manuscript.

Kind regards,

Zulkarnain Jaafar

Academic Editor

PLOS ONE

Journal Requirements:

 “Indian Council of Medical Research, Department of Health Research, Ministry of Health and Family Welfare, Government of India, funded the conduct of the study”

“This study is funded by the Indian Council Medical Research, Department of Health Research, Ministry of Health and Family Welfare, Government of India. We gratefully acknowledge the Director (Education), New Delhi Municipal Council, school principals and teachers for granting the permission and their support for the conduct of the study. We wish to thank all the field data collection and laboratory investigation teams. We wish to gratefully acknowledge Senthil Amudhan, National Institute of Mental Health Sciences, Bengaluru, India. We also wish to thank all the study participants for their support and cooperation for completing this study. We also wish also thank the REDCap administrators for providing user account and its management for storing the study data. “

  “Indian Council of Medical Research, Department of Health Research, Ministry of Health and Family Welfare, Government of India, funded the conduct of the study”

5. In the online submission form, you indicated that [Data cannot be shared publicly because data contains confidential participant information. Data are available on request from the corresponding author.].

Reviewers' comments:

Reviewer's Responses to Questions

**Comments to the Author**

1. Is the manuscript technically sound, and do the data support the conclusions?

Reviewer #1: Yes

Reviewer #2: Yes

2. Has the statistical analysis been performed appropriately and rigorously?

Reviewer #1: Yes

Reviewer #2: Yes

3. Have the authors made all data underlying the findings in their manuscript fully available?

Reviewer #1: Yes

Reviewer #2: No

4. Is the manuscript presented in an intelligible fashion and written in standard English?

Reviewer #1: Yes

Reviewer #2: Yes

5. Review Comments to the Author

Reviewer #1: The manuscript is written in a lucid manner. The authors have addressed the need/importance in a clear elaborate manner. The operational definitions could be attached as an appendix. Sample size could be mentioned within the parantheses () for all table headings.

The manuscript could be "ACCEPTED" for publication.

Reviewer #2: The study describes a technically sound piece of scientific research with data that supports the conclusions. The work is original and addresses an important public health issue in the context of dual burden of disease that the low- and middle-income countries are currently faced with. Comparing two socio-economic groups is a valuable insight, particularly the use of public Vs private as basis for categorizing socio-economic status looks an interesting venture. The methods are well explained and appear replicable. The sample sizes are adequate. The conclusions are drawn appropriately based on the data presented.

The quality of the manuscript could improve substantially if the authors can address some of the issues raised below.

Abstract

In the second sentence of the abstract, the authors state that; “Further, these risks may have aggravated due to worsening food security and diet quality during the pandemic”. The starting word, “Further” could be removed as it adds no value to the meaning of the sentence yet occupies space. Please be specific by ending this statement using ....... during the COVID-19 pandemic.

The statement: “Among 3888 (77.22%) recruited, 1985 (93.02%) were from public schools and 1903 (66.00%) were from private-schools aged 6-19 years” is unclear. What was the denominator, given that 77.22% and 66.00% appear not to have a common denominator? Please revise that sentence and make it clear to the reader.

The authors further state that; “The prevalence of underweight and general obesity was 4.95% (95% CI 1.25-12.72) and 13.41 (95% CI 2.98-33.87) respectively with higher underweight prevalence in public schools (p<0.0001) and obesity prevalence in private-schools (p<0.0001) with adjusted PR=4.42, 95% CI 2.90-6.72)”. Given that the statement refers to prevalence of 2 variables, the word was may not be most suitable. Please replace it with “were”. As differences in prevalence of underweight and obesity were statistically significant, it will be good for the authors to say; ........ respectively with significantly higher underweight prevalence in public schools (p<0.0001) and obesity prevalence in private-schools (p<0.0001) with adjusted PR=4.42, 95% CI 2.90-6.72)”. Please, also be clear what the PR at the end of this sentence represents given the mix in the sentence. The authors might need to consider breaking this long sentence into shorter sentences for clarity.

The sentence; “The association with type of schools was also significant for impaired fasting plasma glucose (adjusted PR=2.37, 95% CI: 1.19-4.72) and metabolic syndrome (adjusted PR=3.51, 95% CI 1.54-8.01)” is not clear. Please revise and make this clear. Please state the type of association in reference and the direction with reference to the independent variable (in this case the type of school).

The authors conclude that; “An effective implementation of food security measures and targeted initiatives will be crucial to mitigate the socio-economic and gender disparities associated with growing burden of cardiometabolic traits”. The reference to gender in this conclusion is not substantiated by results. The authors should consider including some gender differences in the results or remove the reference to gender in this conclusion.

Data Availability

Regarding data availability, the authors state that some restriction may apply regarding data availability. It is important the authors provide a justification as to why data availability is limited. It is now a good research practice to make research data available to readers and other researchers.

The authors add that: “Data cannot be shared publicly because data contains confidential participant information. Data are available on request from the corresponding author”. The authors should consider anonymizing the data before sharing it unless there are unavoidable impediments to anonymizing the data or sharing it.

INTRODUCTION

The introduction is generally well written, apart from a few grammatical errors. The authors should consider a review and revision of grammar by someone whose first language is English for both the abstract and introduction. Adding information on the prevalence of under- and overweight in low- and middle-income countries would give perspective to the background.

METHODS:

It is particularly concerning that the study schools were conveniently sampled and this has an implication on the generalizability of the findings. The authors should consider acknowledging this as a limitation of the study, although the discussion states otherwise.

The “Study setting and recruitment strategy” does not have information on setting. The study setting would for example, explain the total number of schools, number private, then public etc. Such information is missing under this section. The authors should consider including such context or remove ‘study setting’ from the sub-section header.

The ‘was’ in line 157 should be were.

Lines 165 and 166 refer to ethical clearance/study approval under the inclusion and exclusion criteria. Please create a separate sub-section for the ethical consideration and clearance/approval.

The authors have not explained what was done regarding those who did not consent or those discontinued. Were these considered as non-response or replaced? What of those who withdrew after start of the interview/measurements? Please explain what the study did under such circumstances.

Lines 261 and 262: The authors state; “Finally, the total sample size for anthropometry parameters was1980 and for biochemical parameters is 1100 only from public and private schools”. It is not clear if these (1980 and 1100) were independent samples. If so, why didn’t the authors opportunistically consider 1980, the higher of the two as the sample size to cater for both groups?

Ethical consideration: What happened to those detected as underweight, overweight, obese, hypertensive, with low HDL or high LDL etc? Did the study consider managing, referring or providing appropriate counseling as part of ethical responsibilities? It is important that a sub-section is created in the methods section, detailing how such ethical issues were handled, alongside the study approval, that I earlier requested to be separated from inclusion and exclusion criteria.

The authors express using random effects logistic regression models. However, the manuscript does not discuss the model building strategy, and how well the study data fitted the model. Such description would provide confidence in the results provided. Alternatively, the model building strategy and fitting tests can be submitted as supplementary material. Plus, some independent variables have abruptly surfaced in the results, when they were not discussed in the methods. It is important that each dependent variable, and the corresponding independent variables are discussed in the methods so that the reader knows what to expect in the results.

RESULTS

The 95% CI for prevalence of obesity (lines 331-335) are unusually too wide, despite prevalence estimates away from 50%. The authors need to account for this as it puts the precision of estimates to question. The authors should consider merging categories if this (wide 95% CI) is associated with smaller sample sizes across sub-groups. In addition, there appears to be differences between the lower and upper limits of the 95% CIs, with lower values having smaller difference from the estimated prevalence. The authors should provide explanation(s) for such findings.

The concept of early and late adolescents was not explained anywhere in the methods, and suddenly surfaces in the results. Please explain (either in the methods or where those results are presented) the age categories representing these (early and late) sub-groups of adolescents.

Table 2 has ‘Fating’ plasma glucose (mg/dl) instead of fasting .... Please check the different tables and correct this across all.

In Table 2, NCEP ATP-III has unusually very high 95% CI among private school respondents. The authors need to explanation such an anomaly. Plus, it is interesting that some variables have overlap in 95% CI, while the p-values depict significant differences (i.e. p<0.05). The authors should consider explaining this.

Lines 394-395: The authors state that; “The difference in prevalences of dyslipidemia was significantly different among girls (34.90% vs 26.61%, p=0.019) (Table 4)” without specifying the comparison groups. Please make it clear that this is between public and private schools. Use of the description ‘significantly different among girls’ is not clear. Please make it explicit that dyslipidemia prevalence is significantly higher in public than private schools or between girls and boys.

Lines 398-400: “The prevalence of MetS among urban adolescents according to ATP-III criteria and IDF criteria was 3.43% (95% CI 1.04-8.14) and 1.39% (95% CI 0.17-4.87) respectively”. Please specify the gender/sex these specific results refer to.

Lines 400-402: The authors state: “It was significantly lower among public school students compared to private school students (p=0.020) with adjusted PR of 3.51 (95% CI 1.54-8.01) (Table 2 and Fig 3). A similar pattern (Tables 3 and 4) was observed among boys (p=0.022) and girls (0.088)”. Please revise these statements to reflect the gender the results they are referring to.

By stating; “A similar pattern (Tables 3 and 4) was observed among boys (p=0.022) and girls (0.088)”, it is also implied that the difference is statistically significant in both gender’s results. However, the p-value of 0.088 among girls states otherwise. Please revise this.

Regarding Mets, it is also important to present results based on the standards in reference. I am seeing a simultaneous and indiscriminate use of ATP-III criteria and IDF results, and this risks causing confusion in interpretation, let alone raising the sensitivity regarding identification of Mets. It is important to consider results of each standard independently.

DISCUSSION

Line 434: “To our knowledge, this is the first longitudinal school-based study comprehensively assessed the ..........” ‘That’ appears to be missing between study and comprehensively, making the sentence lose meaning.

Line 438: “The findings revealed that the anthropometric, blood pressure,.....”. Should be ‘anthropometric measurements’.

Lines 438-440: The authors state that; “The findings revealed that the anthropometric, blood pressure, lipid levels, fasting plasma glucose and insulin were significantly higher among private school going students (proxy for higher SES) compared to public school students (proxy for lower SES)”. Please review this summary and exclude those measures whose differences were not statistically significant. Plus, it is important to make it clear in the methods section the analysis you considered as final such that discussion will focus on them. As it currently stands, the authors appear to be picking any statistically significant findings across the different analyses. Multi-variate analysis (i.e. the random effects logistic regression) could take precedence as the possible co-founders are controlled for in that (high) level of analysis.

Lines 440-442: The statement; “The prevalence of underweight (nearly five times) was higher in public schools while obesity was higher in private school (more than five times) students as compared to their counterparts” is not clear. Please revise and make it clear to the reader.

Lines 461-462: The authors state that: “These findings will be crucial to understand the adverse impact of COVID-19 on nutrition particularly on the vulnerable and marginalized groups”. Although this study was conducted in the post-COVID era, there is no evidence linking the findings to the COVID-19 pandemic given a lack of baseline pre-pandemic. The authors may consider removing reference to the ‘adverse impacts of COVID-19’ in the discussion. Alternatively, the effect/impact of COVID-19 can be discussed in reference to life-style changes following the lock-downs, not necessarily attributing the observed findings to it as implied here.

Lines 478-479: The authors state: “Similarly, the current study reported lower rates of underweight prevalence among private school children reported previously”. This statement should end with ......... than reported previously. ‘Than’ is missing.

The statement between lines 504-507 ends with a coma (not full stop). Please use a full stop.

Between lines 549 to 551, the authors say the LEAP study will make them establish a causal relationship by assessing determinants of clustering of CVD risks. Although this is a cohort study, it is worth taking note that ‘causal relationship studies’ take on some stringent criteria. The authors should consider revising their choice of the language, ‘causal’ as opposed to an association, unless they are sure the design will deliver a ‘causal relationship’.

Lines 552-554: Please punctuate the sentences therein. Some full stops are being placed within continuing sentences.

GENERAL COMMENTS

Charts/Figures: In their current state, the figures are not clearly visible. The authors should consider submitting high-resolution figures/charts.

The write-up contains several typos and grammatical errors. The authors should consider a review and revision by someone whose first language is English in order to improve the grammar and overall write-up.

6. PLOS authors have the option to publish the peer review history of their article (what does this mean? ). If published, this will include your full peer review and any attached files.

**Do you want your identity to be public for this peer review?** For information about this choice, including consent withdrawal, please see our Privacy Policy .

Reviewer #1: No

Reviewer #2: **Yes: ** Robert Onzima DDM Anguyo

---

## [Author Response · Author response to Decision Letter 1]

3 Mar 2025

To

The Academic Editor

PLOS One

We thank the reviewers for their valuable comments that helped us to improve our manuscript. The following are the point-wise response to reviewers and the line numbers mentioned in the responses are as per the file labeled as ‘Manuscript’.

Response to the Academic Editor

Response: We revised the manuscript as per PLOS ONE's style requirements.

Response: We corrected this also.

“Indian Council of Medical Research, Department of Health Research, Ministry of Health and Family Welfare, Government of India, funded the conduct of the study”

Response: As suggested, we have included the Role of Funder statement in the cover letter.

“This study is funded by the Indian Council Medical Research, Department of Health Research, Ministry of Health and Family Welfare, Government of India. We gratefully acknowledge the Director (Education), New Delhi Municipal Council, school principals and teachers for granting the permission and their support for the conduct of the study. We wish to thank all the field data collection and laboratory investigation teams. We wish to gratefully acknowledge Senthil Amudhan, National Institute of Mental Health Sciences, Bengaluru, India. We also wish to thank all the study participants for their support and cooperation for completing this study. We also wish also thank the REDCap administrators for providing user account and its management for storing the study data. “

“Indian Council of Medical Research, Department of Health Research, Ministry of Health and Family Welfare, Government of India, funded the conduct of the study”

Response: We removed the funding related information from the manuscript.

5. In the online submission form, you indicated that [Data cannot be shared publicly because data contains confidential participant information. Data are available on request from the corresponding author.].

Response: We included the revised statement in the cover letter and updated it in the online submission form also.

Response: We checked the Reference list and edited as necessary. We thank the Academic Editor for all the insights.

Review Comments to the Author:

Reviewer #1: The manuscript is written in a lucid manner. The authors have addressed the need/importance in a clear elaborate manner. The operational definitions could be attached as an appendix. Sample size could be mentioned within the parentheses () for all table headings. The manuscript could be "ACCEPTED" for publication.

Response: We thank the reviewer for the encouraging comments and evaluating our manuscript. The sample size has been mentioned in the heading for Table 1 and for Tables 2-5, it is given inside the tables as it is different for physiological, clinical and biochemical parameters.

Reviewer #2: The study describes a technically sound piece of scientific research with data that supports the conclusions. The work is original and addresses an important public health issue in the context of dual burden of disease that the low- and middle-income countries are currently faced with. Comparing two socio-economic groups is a valuable insight, particularly the use of public Vs private as basis for categorizing socio-economic status looks an interesting venture. The methods are well explained and appear replicable. The sample sizes are adequate. The conclusions are drawn appropriately based on the data presented.

The quality of the manuscript could improve substantially if the authors can address some of the issues raised below.

We thank the reviewer for evaluating our manuscript vividly and helping us to improve the manuscript.

Abstract

Comment 1: In the second sentence of the abstract, the authors state that; “Further, these risks may have aggravated due to worsening food security and diet quality during the pandemic”. The starting word, “Further” could be removed as it adds no value to the meaning of the sentence yet occupies space. Please be specific by ending this statement using ....... during the COVID-19 pandemic.

Response: We thank the reviewer for his suggestion. Lines 26-27 has been edited.

Comment 2: The statement: “Among 3888 (77.22%) recruited, 1985 (93.02%) were from public schools and 1903 (66.00%) were from private-schools aged 6-19 years” is unclear. What was the denominator, given that 77.22% and 66.00% appear not to have a common denominator? Please revise that sentence and make it clear to the reader.

Response: Lines 41-42 has been revised and edited as suggested.

Comment 3: The authors further state that; “The prevalence of underweight and general obesity was 4.95% (95% CI 1.25-12.72) and 13.41 (95% CI 2.98-33.87) respectively with higher underweight prevalence in public schools (p<0.0001) and obesity prevalence in private-schools (p<0.0001) with adjusted PR=4.42, 95% CI 2.90-6.72)”. Given that the statement refers to prevalence of 2 variables, the word was may not be most suitable. Please replace it with “were”. As differences in prevalence of underweight and obesity were statistically significant, it will be good for the authors to say; ........ respectively with significantly higher underweight prevalence in public schools (p<0.0001) and obesity prevalence in private-schools (p<0.0001) with adjusted PR=4.42, 95% CI 2.90-6.72)”. Please, also be clear what the PR at the end of this sentence represents given the mix in the sentence. The authors might need to consider breaking this long sentence into shorter sentences for clarity.

Response: We thank the reviewer for bringing the clarity in the abstract results. Lines 42-48 has been revised.

Comment 4: The sentence; “The association with type of schools was also significant for impaired fasting plasma glucose (adjusted PR=2.37, 95% CI: 1.19-4.72) and metabolic syndrome (adjusted PR=3.51, 95% CI 1.54-8.01)” is not clear. Please revise and make this clear. Please state the type of association in reference and the direction with reference to the independent variable (in this case the type of school).

Response: Lines 50-52 has been revised as suggested.

Comment 5: The authors conclude that; “An effective implementation of food security measures and targeted initiatives will be crucial to mitigate the socio-economic and gender disparities associated with growing burden of cardiometabolic traits”. The reference to gender in this conclusion is not substantiated by results. The authors should consider including some gender differences in the results or remove the reference to gender in this conclusion.

Response: We agree with the reviewer in this context and Line 64 has been edited as suggested.

Data Availability:

Comment 6: Regarding data availability, the authors state that some restriction may apply regarding data availability. It is important the authors provide a justification as to why data availability is limited. It is now a good research practice to make research data available to readers and other researchers. The authors add that: “Data cannot be shared publicly because data contains confidential participant information. Data are available on request from the corresponding author”. The authors should consider anonymizing the data before sharing it unless there are unavoidable impediments to anonymizing the data or sharing it.

Response: We thank the reviewer for his valuable input and we agree that making data available to readers and researchers is immensely important and a valuable practice. In this manuscript, we have provided all summary data using baseline data of the ongoing longitudinal study either in the manuscript itself or as supplementary information. Since the study is still in progress, the disclosure of raw data will be subject to the policies of the funding agency and the authors' institution.

INTRODUCTION

Comment 7: The introduction is generally well written, apart from a few grammatical errors. The authors should consider a review and revision of grammar by someone whose first language is English for both the abstract and introduction. Adding information on the prevalence of under- and overweight in low- and middle-income countries would give perspective to the background.

Response: We appreciate and value the comments of the reviewer. We have added the prevalences of underweight, overweight and obesity in low-and middle-income countries (Lines 75-77) in the introduction. We also revised the Abstract and Introduction as suggested.

METHODS:

Comment 8: It is particularly concerning that the study schools were conveniently sampled and this has an implication on the generalizability of the findings. The authors should consider acknowledging this as a limitation of the study, although the discussion states otherwise.

Response: As suggested, we have added this as a limitation and revised statement in the discussion (Lines 475-476).

Comment 9: The “Study setting and recruitment strategy” does not have information on setting. The study setting would for example, explain the total number of schools, number private, then public etc. Such information is missing under this section. The authors should consider including such context or remove ‘study setting’ from the sub-section header. The ‘was’ in line 157 should be were.

Response: We thank the reviewer for his suggestion. Lines 141-145 and 156 has been revised and edited.

Comment 10: Lines 165 and 166 refer to ethical clearance/study approval under the inclusion and exclusion criteria. Please create a separate sub-section for the ethical consideration and clearance/approval. The authors have not explained what was done regarding those who did not consent or those discontinued. Were these considered as non-response or replaced? What of those who withdrew after start of the interview/measurements? Please explain what the study did under such circumstances.

Response: We thank the reviewer for his valuable comment. Ethical clearance/study approval has been added as a separate sub-section titled, ‘Ethical consideration and approval’ in Lines 165-174. Lines 175-181 provide the criteria for non-respondents and those who discontinued. Out of 5045 eligible students, 3888 (77.07%) participated in the survey (private =1903 and public=1985). The non-response rate in private and public schools were 34.45% and 6.98% respectively. These non-respondents include students who did not get consent, consented but absent and did not give assent (after start of the interview). To increase the response rate, multiple reminders were sent to those students’ parents by email/WhatsApp class group ids and a study representative conducted sessions with the parents during parents-teacher meeting also. Non-response was higher among private school students than public school students, primarily due to parental hesitance about medical procedures like blood sampling in a school setting. This reluctance, particularly among parents from higher socioeconomic backgrounds, was heightened by the unprecedented circumstances of the COVID-19 pandemic. We added this in the discussion in Lines 476-482.

Comment 11: Lines 261 and 262: The authors state; “Finally, the total sample size for anthropometry parameters was1980 and for biochemical parameters is 1100 only from public and private schools”. It is not clear if these (1980 and 1100) were independent samples. If so, why didn’t the authors opportunistically consider 1980, the higher of the two as the sample size to cater for both groups?

Response: The focus of the study for the blood pressure/biochemical parameters were to include a subset of students aged 11-15 years from the total sample size of 1980. This selection was made because the focus of the study was only among 11–15-year age group for these parameters. It has now been clarified/corrected in the manuscript; please refer to Lines 245-248.

Comment 12: Ethical consideration: What happened to those detected as underweight, overweight, obese, hypertensive, with low HDL or high LDL etc? Did the study consider managing, referring or providing appropriate counselling as part of ethical responsibilities? It is important that a sub-section is created in the methods section, detailing how such ethical issues were handled, alongside the study approval, that I earlier requested to be separated from inclusion and exclusion criteria.

Response: We appreciate the reviewer’s insightful suggestion. Individualized measurement reports, sealed in envelopes, were distributed to all participants through their schools. The list of participants with abnormal findings was provided to the respective school principals, who were asked to inform parents during the parent-teacher meeting. During this meeting, we requested the teachers to highlight the importance of physical activity, a balanced diet, and reduced screen time to support the children's well-being. The team had no direct role in counselling, as the school authorities did not permit them to interact with the participants' parents. However, the team offered medical support to parents who requested assistance for their children. Lines 165-174 has been revised and edited accordingly.

Comment 13: The authors express using random effects logistic

---

## [Decision Letter · Decision Letter 1]

9 Mar 2025

PONE-D-24-53240R1Cardio-metabolic traits and its socioeconomic differentials among school children including metabolically obese normal weight phenotypes in India: A post-COVID baseline characteristics of LEAP-C cohortPLOS ONE

Dear Dr. Mani,

Thank you for submitting your manuscript to PLOS ONE. After careful consideration, we feel that it has merit but does not fully meet PLOS ONE’s publication criteria as it currently stands. Therefore, we invite you to submit a revised version of the manuscript that addresses the points raised during the review process.

**ACADEMIC EDITOR: Dear Author, please make necessary changes to the manuscript as suggested by the reviewer.**==============================

We look forward to receiving your revised manuscript.

Kind regards,

Zulkarnain Jaafar

Academic Editor

PLOS ONE

Journal Requirements:

Reviewers' comments:

Reviewer's Responses to Questions

**Comments to the Author**

1. If the authors have adequately addressed your comments raised in a previous round of review and you feel that this manuscript is now acceptable for publication, you may indicate that here to bypass the “Comments to the Author” section, enter your conflict of interest statement in the “Confidential to Editor” section, and submit your "Accept" recommendation.

Reviewer #2: (No Response)

2. Is the manuscript technically sound, and do the data support the conclusions?

Reviewer #2: Yes

3. Has the statistical analysis been performed appropriately and rigorously?

Reviewer #2: Yes

4. Have the authors made all data underlying the findings in their manuscript fully available?

Reviewer #2: No

5. Is the manuscript presented in an intelligible fashion and written in standard English?

Reviewer #2: Yes

6. Review Comments to the Author

Reviewer #2: The data for this study is not publicly available. This is understandable given that the data used in this study/analysis is an extract of data from an on-going study for which raw data may not be made public. It is good to note that the authors accept to make the study data available to individuals who request for them.

Lines 140-141: Under the study setting and recruitment strategy, the authors state that; “The study setting comprised five schools in Delhi, India, including three public and two private schools (convenient selection as schools were reluctant due to the pandemic).” Please note that this does not describe the ‘study setting’. I guess the five schools (three public and two private) is the sample size of schools. Study setting means the physical, social, and cultural context where the research is conducted. In the case of this study, the setting would (for example) mean describing the total number of schools in Delhi. Out of the total schools, what number are private and public. What is the proportion of the population going to private Vs public schools (if such data exists). Then it is also good context to add that in India (Delhi in this case), it is general practice that children from wealthy or higher socioeconomic groups generally go to private schools, while those from the poorer or lower socioeconomic groups go to public schools, citing appropriate sources where applicable. This would amount to setting.

Please distinguish your sample from study setting because the setting refers to the natural context (set-up) in the study area. As I stated in my initial review, if you cannot add the context, please consider removing ‘Study setting’ from the title “Study setting and recruitment strategy”. Recruitment strategy should suffice in this case. Please, also remove ‘setting’ from description of the sample as in line 140.

7. PLOS authors have the option to publish the peer review history of their article (what does this mean? ). If published, this will include your full peer review and any attached files.

**Do you want your identity to be public for this peer review?** For information about this choice, including consent withdrawal, please see our Privacy Policy .

Reviewer #2: **Yes: ** Robert Anguyo Onzima

---

## [Author Response · Author response to Decision Letter 2]

11 Mar 2025

Response to Reviewer

We thank the reviewers for their valuable comments that helped us to improve our manuscript. The following are the point-wise response to reviewers and the line numbers mentioned in the responses are as per the file labeled as ‘Manuscript’.

Review Comments to the Author

Reviewer #2: The data for this study is not publicly available. This is understandable given that the data used in this study/analysis is an extract of data from an on-going study for which raw data may not be made public. It is good to note that the authors accept to make the study data available to individuals who request for them.

Response: We agree with the reviewer. We are uploading the following for Data Availability: “The data used in this study is an extract from an ongoing study and is not currently publicly available. The data that support the findings of this study are available on reasonable request from the corresponding author subject to data sharing policy and ethical restrictions of the institute”.

Lines 140-141: Under the study setting and recruitment strategy, the authors state that; “The study setting comprised five schools in Delhi, India, including three public and two private schools (convenient selection as schools were reluctant due to the pandemic).” Please note that this does not describe the ‘study setting’. I guess the five schools (three public and two private) is the sample size of schools. Study setting means the physical, social, and cultural context where the research is conducted. In the case of this study, the setting would (for example) mean describing the total number of schools in Delhi. Out of the total schools, what number are private and public. What is the proportion of the population going to private Vs public schools (if such data exists). Then it is also good context to add that in India (Delhi in this case), it is general practice that children from wealthy or higher socioeconomic groups generally go to private schools, while those from the poorer or lower socioeconomic groups go to public schools, citing appropriate sources where applicable. This would amount to setting.

Please distinguish your sample from study setting because the setting refers to the natural context (set-up) in the study area. As I stated in my initial review, if you cannot add the context, please consider removing ‘Study setting’ from the title “Study setting and recruitment strategy”. Recruitment strategy should suffice in this case. Please, also remove ‘setting’ from description of the sample as in line 140.

Response: I thank the reviewer and completely agree with him. As suggested by the reviewer, we have removed ‘Study setting’ from the title “Study setting and recruitment strategy” and also “setting” from line 140.

---

## [Editor Report · Decision Letter 2]

14 Mar 2025

Cardio-metabolic traits and its socioeconomic differentials among school children including metabolically obese normal weight phenotypes in India: A post-COVID baseline characteristics of LEAP-C cohort

PONE-D-24-53240R2

Dear Dr. Mani,

We’re pleased to inform you that your manuscript has been judged scientifically suitable for publication and will be formally accepted for publication once it meets all outstanding technical requirements.

Kind regards,

Zulkarnain Jaafar

Academic Editor

PLOS ONE
---

## [Editor Report · Acceptance letter]

PONE-D-24-53240R2

PLOS ONE

Dear Dr. Mani,

I'm pleased to inform you that your manuscript has been deemed suitable for publication in PLOS ONE. Congratulations! Your manuscript is now being handed over to our production team.

Kind regards,

on behalf of

Dr. Zulkarnain Jaafar

Academic Editor

PLOS ONE